# Serotonin signaling by maternal neurons upon stress ensures progeny survival

Srijit Das[1], Felicia K Ooi[1], Johnny Cruz Corchado[1], Leah C Fuller[2], Joshua A Weiner[2,3], Veena Prahlad[1,2,3]*

[1]Department of Biology, Aging Mind and Brain Initiative, Iowa City, United States; [2]Department of Biology, Iowa City, United States; [3]Iowa Neuroscience Institute, Iowa City, United States

**Abstract** Germ cells are vulnerable to stress. Therefore, how organisms protect their future progeny from damage in a fluctuating environment is a fundamental question in biology. We show that in *Caenorhabditis elegans*, serotonin released by maternal neurons during stress ensures the viability and stress resilience of future offspring. Serotonin acts through a signal transduction pathway conserved between *C. elegans* and mammalian cells to enable the transcription factor HSF1 to alter chromatin in soon-to-be fertilized germ cells by recruiting the histone chaperone FACT, displacing histones, and initiating protective gene expression. Without serotonin release by maternal neurons, FACT is not recruited by HSF1 in germ cells, transcription occurs but is delayed, and progeny of stressed *C. elegans* mothers fail to complete development. These studies uncover a novel mechanism by which stress sensing by neurons is coupled to transcription response times of germ cells to protect future offspring.

## Introduction

The ability to react rapidly to environmental challenges is critical for the survival of individuals and species. In organisms with a nervous system, sensory neuronal circuits initiate many of the animals' responses to environmental stressors, modifying behavior and physiology to adapt to the altered circumstance. However, whether and how sensory information used by the organism to predict impending danger is coupled to the protection of its future offspring is largely unknown. One conserved signaling molecule that is released in most organisms, including *C. elegans*, early in response to real or perceived threats is the neuromodulator serotonin (5-hydroxytryptamine, 5-HT) (*Carhart-Harris and Nutt, 2017*; *Chaouloff, 2002*; *Chaouloff et al., 1999*; *Deakin, 2013*; *Faulkner and Deakin, 2014*; *Ooi and Prahlad, 2017*; *Tatum et al., 2015*). 5-HT is a bioamine secreted by specific neurons, and in some cases by peripheral cells, to modify learning and memory, behavior, development and physiological processes (*Berger et al., 2009*; *Carhart-Harris and Nutt, 2017*; *Chaouloff, 2002*; *Chaouloff et al., 1999*; *Deakin, 2013*; *Faulkner and Deakin, 2014*; *Ooi and Prahlad, 2017*; *Tatum et al., 2015*), facilitating the animals' future response to the stressor. For instance, in *Aplysia*, 5-HT increase mediates the encoding of memory required for habituation to a specific stressor and non-associative learning (*Kandel and Schwartz, 1982*). In mammals, increased 5-HT plays a dominant role in learning following social stress (*Bauer, 2015*; *Chaouloff et al., 1999*). In *C. elegans*, we and others have shown that enhanced 5-HT mediates learned avoidance and activates defense responses. For example, pathogen odors increase 5-HT levels in *C. elegans* through the activity of chemosensory neurons (*Bayer and Hobert, 2018*; *Chao et al., 2004*; *Song et al., 2013*; *Zhang et al., 2005*), and this increase in 5-HT is required for both the animal's subsequent avoidance of pathogens, and its protection from infection. Similarly, exposure to increasing temperatures enhances 5-HT release from the serotonergic neurons (called NSM and ADF neurons) through the activity of the animal's thermosensory neurons (called AFD neurons) and this release of 5-HT cell

*For correspondence:
veena-prahlad@uiowa.edu

**Competing interests:** The authors declare that no competing interests exist.

non-autonomously protects the animal from proteotoxicity (*Gracida et al., 2017*; *Iwanir et al., 2016*; *Shao et al., 2019*; *Tatum et al., 2015*). However, whether 5-HT released by the parent upon the detection of stress protects germ cells and future progeny from stress is not known. In fact, mammalian studies are suggestive of the opposite role for elevated levels of 5-HT that accompany chronic stress in the parent, and increased 5-HT is thought to contribute to behavioral and psychiatric disorders such as schizophrenia, depression, and autism in progeny through as yet poorly understood mechanisms (*Bonnin et al., 2011*; *Dowell et al., 2019*; *McKay, 2011*; *Shah et al., 2018*)—an unexpected effect given that stress-induced release of 5-HT by neurons and other 5-HT synthesizing cells is a highly conserved phenomenon.

Here, we asked whether the stress-induced release of 5-HT by maternal neurons provides any benefits to germ cells and the development of future progeny. We used *C. elegans* to address this question in an *in vivo* setting, and cultured mammalian cells to dissect the molecular pathways by which 5-HT might act and to examine the extent to which 5-HT-mediated effects are conserved. We show that in *C. elegans*, 5-HT released by maternal neurons upon stress allows the transcription factor heat shock factor 1 (HSF1) to shorten the time to onset of mRNA production in soon-to-be fertilized germ cells. Specifically, 5-HT promotes the post-translational modification of HSF1 by protein kinase A (PKA) allowing HSF1 to recruit the histone chaperone FACT (FAcilitates Chromatin Transcription) and alter histone dynamics to initiate transcription. This timely activation of HSF1 in germ cells ensures their viability and future stress tolerance: embryos that arise from heat-shocked mothers contain an excess of protective mRNA and are more resilient to subsequent temperature insults as larvae. In the absence of maternal 5-HT, HSF1 activation in the germline is delayed, occurs without the recruitment of FACT, and a large fraction of embryos derived from these germ cells do not complete development, nor do they exhibit transgenerational thermotolerance. Remarkably, the intracellular signal transduction pathway by which 5-HT enables HSF1 to recruit FACT is conserved between *C. elegans* and mammalian cells. These results provide a novel mechanism by which 5-HT signaling protects germ cells, and developmental integrity. In addition, they elucidate a molecular mechanism by which transcription response times of specific cells in a metazoan are tuned to stimulus intensity and onset.

## Results

### Maternal serotonin protects the germline from the detrimental effects of temperature stress

In *C. elegans* the only source of 5-HT is neuronal (*Sze et al., 2000*). Tryptophan hydroxylase, TPH-1, the rate-limiting enzyme for 5-HT synthesis, is expressed only in serotonergic neurons of hermaphrodites, and 5-HT synthesized and released by these neurons not only modifies neural circuit activity but is also distributed throughout the animal *via* the coelomic fluid to act on 5-HT receptors expressed by peripheral tissues (*Chase and Koelle, 2007*; *Curran and Chalasani, 2012*; *Tierney, 2001*). A deletion mutant in *tph-1* is viable and grossly wild-type, although completely devoid of 5-HT, and therefore deficient in all responses that require 5-HT (*Sze et al., 2000*). Therefore, to examine whether 5-HT released by maternal neurons upon the sensing of stress affected germ cells, we exposed wild-type animals and *tph-1* mutant animals to a transient and brief temperature-stress that we had previously shown enhances 5-HT release (5 min exposure to 34°C; a temperature gradient of 1°C ± 0.2°C increase per minute; see Materials and methods), and then, evaluated the survival and development of their embryos laid post-heat shock (*Figure 1—figure supplement 1A*). Examining the numbers of already-fertilized embryos, the partially cellularized oocytes in adult animals that were soon-to-be fertilized, and the numbers of embryos laid during, and following heat exposure allowed us to extrapolate that embryos laid 0–2 hr following heat treatment were the already-fertilized embryos present *in utero* when the mothers were subjected to heat shock, and those generated between 2 and 4 hr following maternal heat-shock would be generated from partially cellularized germ cell nuclei (oocytes) present in the syncytial germline during the transient heat shock (*Figure 1—figure supplement 1B–D*). This was true for both wild-type and *tph-1* mutant animals, although consistent with previously published data, 5-HT is required to modulate egg laying rates (*Chase and Koelle, 2007*) and control *tph-1* mutants laid variable numbers of eggs during a given 2 hr interval (*Figure 1—figure supplement 1C,D*).

In the absence of stress, all the embryos laid by wild-type as well as *tph-1* mutant animals hatched and developed into gravid adults indicating that 5-HT is not required for the survival of embryos under normal growth conditions (*Figure 1—figure supplement 1E,F* and *Figure 1A*). Wild-type and *tph*-1 gravid adults themselves also survived the 5-min heat exposure with no visible signs of damage (n = 46 experiments, 4–5 animals/experiment; % survival wild-type and *tph-1* mutant animals = 100), consistent with previous reports (*Kourtis et al., 2012*; *Kumsta and Hansen, 2017*; *Ooi and Prahlad, 2017*; *Wang et al., 2018*) that exposure to high temperatures (35–37°C) for longer durations (hours) is required to impact the survival of adult *C. elegans*. However, the brief exposure to temperatures of 34°C was enough to disrupt embryonic development and ~50% of embryos failed to hatch (*Figure 1—figure supplement 1E,F*). This was the case for embryos from wild-type or *tph-1* mutant mothers, irrespective of whether they were present *in utero* when the parents were subjected to the 5-min heat shock (*Figure 1—figure supplement 1F*), or whether the embryos were first laid and then subjected to a 5 min heat-shock (*Figure 1—figure supplement 1E*). Thus, it appeared that development processes were extraordinarily vulnerable to heat-induced disruption.

While the already-fertilized embryos of wild-type and *tph-1* mutant animals were susceptible to heat-shock, this was not the case for embryos derived from the fertilization of partially cellularized germ cells that were fertilized following heat shock of the parents and laid between 2 and 4 hr post-heat shock. These embryos survived and developed into adults—but only if they were derived from wild-type animals, and not from *tph-1* mutant animals (*Figure 1A*). Thus, while almost all the embryos (94 ± 2%; n = 28 experiments, embryos from 4 to 5 animals/experiment) generated from germ cells resident in wild-type animals during heat shock, but fertilized subsequently, hatched to develop into gravid adults (*Figure 1A*), only approximately 50% of the embryos generated similarly by germ cells in *tph-1* animals hatched (53 ± 2%; n = 28 experiments, embryos from 4 to 5 animals/ experiment; *Figure 1A*). This was surprising given the transient nature of the heat exposure and the fact that the germ cells were being fertilized and laid 2 hr following heat shock of the parents, at normal growth temperatures. Exposure of *C. elegans* to exogenous 5-HT causes 5-HT to be taken up into the serotonergic neurons and subsequently released, mimicking endogenous 5-HT (*Cruz-Corchado et al., 2020*; *Jafari et al., 2011*), although the kinetics of uptake and amounts released are not known. We tested whether such a treatment could rescue lethality in the *tph-1* mutant embryos fertilized post-heat shock. Indeed, exposure of *tph-1* mutant mothers to exogenous 5-HT for even only 2 hr (during the 5 min heat-shock treatment and the 2 hr until egg laying) rescued, in a significant manner, the lethality caused by the 5 min heat-shock (*Figure 1—figure supplement 1G*). These data suggested that the presence of 5-HT protected the soon-to-be-fertilized germ cells from transient temperature fluctuations.

To investigate how 5-HT protected germ cells we asked whether the source of 5-HT that rescued embryonic lethality was maternal or embryonic, and whether it acted through the sperm or the female germline. To distinguish between a maternal and potentially embryonic sources, we examined the fate of embryos that were heterozygous for the *tph-1* mutant allele and therefore capable of synthesizing 5-HT, when laid by stressed *tph-1* mutant mothers devoid of 5-HT. If maternal 5-HT was responsible for the protective effects, these heterozygous embryos should remain equally susceptible to heat stress, despite being able to synthesize 5-HT. This was the case and embryos heterozygous for the *tph-1* mutant allele, generated by mating wild-type males with *tph-1* mutant hermaphrodites, were equally susceptible to the 5 min heat-shock as *tph-1* homozygous embryos laid by stressed *tph-1* mutant mothers (*Figure 1B*). Thus, it appeared that 5-HT required for the viability of the germ cells was of maternal origin.

Since embryos of wild-type animals were susceptible to heat if they were fertilized prior to the heat-shock, but were resistant if they were fertilized following heat-shock, we asked whether survival was conferred by some sperm-derived factors generated by the 5-min heat exposure. To answer this, we assessed whether the embryos derived from oocytes fertilized by sperm from heat-shocked males survived the 5-min heat shock. We verified that the embryos being assessed were indeed cross-fertilized with the heat-shocked sperm, and not self-fertilized by non-heat shocked sperm, by determining the sex ratios of the embryos laid by these mated mothers (see Materials and methods). Irrespective of whether the embryos were generated by oocytes fertilized by heat-shocked sperm or by oocytes fertilized by non-heat shocked sperm, ~50% of the embryos did not hatch if they were present *in utero* in mothers subjected to the 5-min heat-shock (*Figure 1C*).

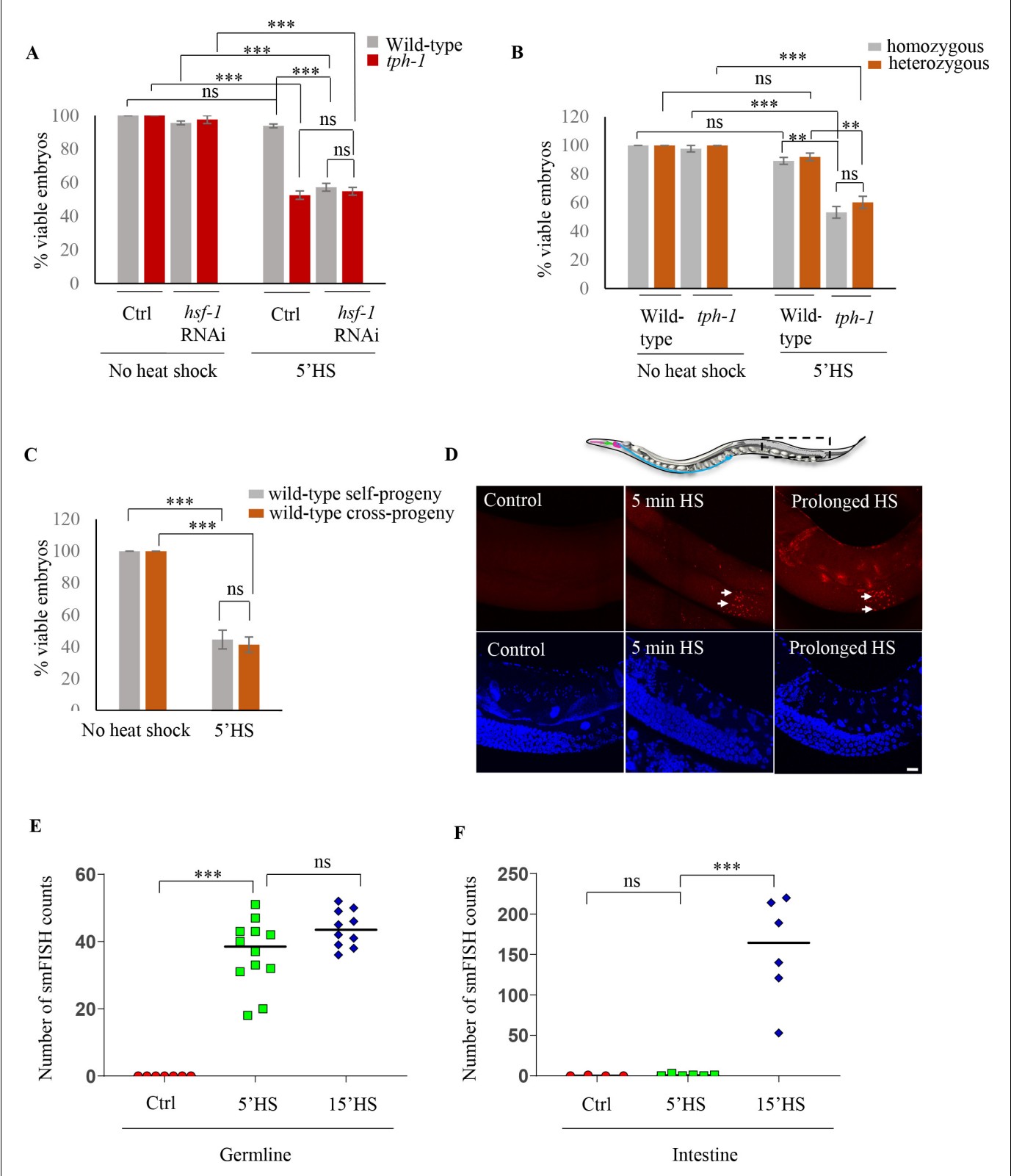

**Figure 1.** Maternal serotonin protects the viability of offspring following heat shock. (**A**) Percent viable embryos from control (No heat-shock) and heat-shocked (5'HS) wild-type and *tph-1* mutant animals under control (Ctrl) conditions and subjected to *hsf-1* RNAi. Embryos were laid during a 2 hr interval by non-heat shocked animals, or animals that were heat-shocked for 5 min at 34°C and allowed to recover at 20°C for 2 hr. Wild-type animals (n = 28 experiments, embryos from 4 to 5 animals/experiment), *tph-1* mutant animals (n = 28 experiments, embryos from 4 to 5 animals/experiment), wild-type

*Figure 1 continued on next page*

Figure 1 continued

*hsf-1* RNAi-treated animals (n = 8 control and 24 heat-shock experiments, embryos from 4 to 5 animals/experiment), and *tph-1 hsf-1* RNAi-treated animals (n = 3 control and 9 heat-shock experiments, embryos from 4 to 5 animals/experiment). Percent viable embryos from control non-heat shocked animals on OP50 are shown here and do not differ from those from animals on Control RNAi (n = 3–8 experiments; control wild-type = 99.5 ± 0.4, control *tph-1* = 98.8 ± 0.6). (B) Percent viable homozygous or heterozygous embryos from control (No heat shock) and heat-shocked (5'HS) wild-type and *tph-1* mutant animals. Wild-type and *tph-1* mutant hermaphrodites were allowed to mate with wild-type males and once the hermaphrodites were laying cross- progeny, embryos laid during a 2 hr interval by non-heat-shocked animals, or animals that had recovered at 20°C for 2 hr post-5 minutes 34°C heat-shock were scored. n = 4 experiments, embryos from 4 to 5 animals/experiment. (C) Percent viable embryos that were either self- or cross-progeny, laid by control (No heat shock) wild-type animals, or wild-type animals heat-shocked for 5 min at 34°C (5'HS). Wild-type hermaphrodites were allowed to mate with wild-type males that had been heat-shocked for 5 min at 34°C, and once the hermaphrodites were laying cross-progeny, the hermaphrodites were heat-shocked for 5 min at 34°C, and viable embryos laid 0–2 hr post-heat shock were scored (n = 5 experiments, embryos from 4 to 5 animals/experiment). (D) Representative confocal image showing *hsp70* (*F44E5.4/.5*) mRNA localization using smFISH in wild-type animals under control conditions and upon 5 min and 15 min (prolonged HS) exposure to 34°C (n = 3 experiments, 24 animals). Optical sections were projected on one plane. Top, red: *hsp70* (*F44E5.4/.5*) mRNA. Bottom, blue: DAPI staining nuclei. Arrows indicate *hsp* mRNA in germline cells and arrowhead, in intestinal cells. Scale bar = 10 µm. (E) smFISH counts in the late pachytene area of the germline of control animals (Ctrl; mean count = 0 ± 0, n = 7), animals subjected to 5 min heat shock (5'HS; mean count = 36.4 ± 2.9, n = 12), and animals subjected to 15 min heat shock (15'HS; mean count = 43.8 ± 1.7, n = 10). (F) smFISH counts in the intestine of control animals (Ctrl; mean count = 0.25 ± 0.25, n = 4), animals subjected to 5 min heat shock (5'HS; mean count = 0.83 ± 0.48, n = 6), and animals subjected to 15 min heat shock (15'HS; mean count = 156.2 ± 26.2, n = 6). Data in A, B, C, show Mean ± Standard Error of the Mean. Data in E, F) show individual values. ***, p<0.001; **p<0.01, *p<0.05 (paired Student's t-test). ns, non-significant.

The online version of this article includes the following figure supplement(s) for figure 1:

**Figure supplement 1.** Characterizing the effects of heat on viability of embryos.
**Figure supplement 2.** A brief (5 min) heat-shock induces HSF-1-dependent gene expression.
**Figure supplement 3.** Characterization of *hsp70* (*F44E5.4/.5*) gene expression upon heat shock.

These data, together, showed that embryonic development was easily disrupted by temperature fluctuations, and maternal 5-HT protected the soon-to-be fertilized germ cells from stress-induced disruption, ensuring their survival. Furthermore, these data indicated that the effects of maternal 5-HT occurred, directly or indirectly, on the partially cellularized female germ cells of the parent hermaphrodite.

## Activation of HSF-1 in the germline is required to protect soon-to-be fertilized germ cells from heat stress

In all cells and organisms, a conserved and essential transcriptional response, the so-called 'heat shock response' counteracts the detrimental effects of heat or other stressors through the activation of the stress-inducible transcription factor, 'heat shock factor 1' (HSF1) (*Anckar and Sistonen, 2011*; *Gomez-Pastor et al., 2018*; *Li et al., 2017*; *Vihervaara et al., 2018*). This transcriptional response of HSF-1 (the sole *C. elegans* HSF1) was essential for the protection of germ cells upon heat exposure as, similar to *tph-1* mutant animals, nearly half the embryos (43 ± 4% n = 15 experiments, embryos from 4 to 5 animals/experiment; *Figure 1A*) generated from germ cells resident in mothers subjected to RNA interference (RNAi)-induced knock-down of HSF-1 did not hatch when the mothers were subjected to 5 min of temperature stress. In contrast, almost all embryos (93.7 ± 1%, n = 3 experiments; embryos from 4 to 5 animals/experiment; *Figure 1A*) laid by *hsf-1* downregulated animals hatched and grew into mature adults in the absence of heat-shock showing that it was the heat-induced activity of HSF-1 in the parent, and not its basal role, that was required to protect germ cells. The adults with downregulated *hsf-1* themselves survived the 5 min heat-shock with no visible defects (n = 15 experiments, 4–5 animals/experiment). In addition, downregulation of *hsf-1* did not further exacerbate the effects of the loss of 5-HT on progeny survival, and 54.9 ± 2.3% of embryos laid by *tph-1* mutant animals subjected to *hsf-1* RNAi also did not hatch, suggesting that the two acted in the same pathway (n = 9 experiments; embryos from 4 to 5 animals/experiment; *Figure 1A*).

In agreement with the requirement for HSF-1 in stress-induced protection of embryos, a 5 min exposure to heat was sufficient to activate HSF-1 and increase the expression of 408 genes, enriched in Biological Processes that handle misfolded proteins, at 0.01 FDR (*Figure 1—figure supplement 2A–C*; *Supplementary file 1*). All these genes were, either directly or indirectly, dependent on HSF-1 (*Hajdu-Cronin et al., 2004*) as a mutant that lacked the trans-activation domain of HSF1,

previously shown to be viable but incapable of eliciting heat-induced transcriptional changes, showed no changes in gene expression upon heat shock (0.01 FDR; *Figure 1—figure supplement 2B*; *Supplementary file 2*). The differentially expressed genes included the major stress-induced *hsp70* genes, *hsp70* (*F44E5.4/.5*) and *hsp70* (*C12C8.1*), as well as other molecular chaperone genes that are the main targets of HSF-1 (*Li et al., 2017*; *Prahlad et al., 2008*) and are known to counter-act heat-induced damage (*Figure 1—figure supplement 2C*; *Supplementary file 1*).

Although the HSF-1-dependent transcriptional response upon heat shock has been well studied in *C. elegans*, its tissue-specificity within this metazoan is unclear. Importantly, it is not known whether germline cells express protective HSF-1 targets to account for their protection. Therefore, to examine whether HSF-1 was indeed activated in the germline upon the 5 min of heat-shock, we localized *hsp70* (*F44E5.4/.5*) mRNA in the whole animals following a 5 min heat-shock using small molecule fluorescent *in situ* hybridization (smFISH). A 5 min exposure to heat stress induced *hsp70* (*F44E5.4/.5*) mRNA predominantly in germline cells in late meiotic prophase and in very few other cells of the animal (*Figure 1D–F*; *Figure 1—figure supplement 3A,B*). These data suggested that HSF-1 targets were indeed activated in germline cells following a 5 min heat-shock. Moreover, the germline was also amongst the first tissues to induce *hsp70* (*F44E5.4/.5*) mRNA, and mRNA was visible in germ cells after 5 min of heat-shock by smFISH, whereas continued exposure to heat was required to detect *hsp70* (*F44E5.4/.5*) mRNA in other cells of the animal (*Figure 1D–F*, *Figure 1—figure supplement 3A,B*).

To confirm that *hsp* genes were indeed expressed in the germline upon 5 min of heat-shock, we utilized two additional methods. First, we knocked-down *hsf-1* in germline tissue using two different strains known to largely restrict all feeding-induced RNAi-mediated gene knock-down to germline tissue: (*mkcSi13* [*sun-1p::rde-1::sun-1 3′UTR + unc-119*(+)] *II; rde-1*(*mkc36*) *V*) (*Zou et al., 2019*) and *rrf-1* (*pk1417*) (*Sijen et al., 2001*) mutants. We then examined the levels of *hsp70* (*F44E5.4/.5*) induced following a 5 min heat shock. We used two independent mutant backgrounds because tissue-specific RNAi in *C. elegans*, especially in the *rrf-1* mutants, has been shown to be leaky and can also occur in intestinal and epithelial cells (*Kumsta and Hansen, 2012*). Second, we used a strain that harbors a temperature sensitive mutation in *glp-4* and fails to develop a germline at the restrictive temperature of 25°C (but has a fully functional germline at 15°C) and examined its transcriptional response to a 5 min exposure to heat. This allowed us to assess the contribution of the germline to the total amount of *hsp70* (*F44E5.4/.5*) mRNA produced. Decreasing the levels of HSF-1 in the germline using the tissue-specific RNAi strains led to a marked decrease in the levels of *hsp70* (*F44E5.4/.5*) mRNA following a 5 min heat-shock (*Figure 2A,B*). In contrast, the knock-down of *hsf-1* in intestinal cells, using a related tissue-specific RNAi strain (*rde-1*(*ne219*) *V; kbIs7*) did not significantly change the levels of *hsp70* (*F44E5.4/.5*) mRNA induced upon the 5 min heat stress (*Figure 2C*). Similarly, abolishing the presence of germline cells using *glp-4* mutant animals grown at the restrictive temperature of 25°C dramatically decreased *hsp70* (*F44E5.4/.5*) levels following a 5 min heat-shock when compared to *glp-4* mutant animals that possessed functional germlines grown at permissive temperatures (15°C) (*Figure 2D*). Moreover, *glp-4* mutant animals that possessed functional germlines because they were grown at permissive temperatures (15°C) induced similar levels of *hsp70* (*F44E5.4/.5*) mRNA as wild-type animals that were grown at 15°C (*Figure 2D*), confirming that these results were not a mere consequence of a change in cultivation temperature. These data, together, indicated that the majority of *hsp70* mRNA produced by wild-type animals following the 5 min heat shock was produced by germline cells.

Examination of the mRNA content of embryos laid by wild-type animals 2–4 hr after they had undergone a 5-min heat shock revealed that these embryos had increased levels of *hsp70* (*F44E5.4/.5*) and *hsp70* (*C12C8.1*) mRNA compared to embryos from control parents, perhaps accounting for their ability to survive the detrimental effects of heat (*Figure 2E,F*). In addition, the larvae that arose from these embryos displayed transgenerational protection from prolonged heat stress (*Figure 2G*). Specifically, when larvae were subjected to a 3 hr exposure to 34°C, a condition titrated to achieve ~50% lethality amongst progeny of control, non-heat-shocked animals, significantly more larvae survived if they were progeny of heat-shocked mothers than if they were offspring of animals grown at control conditions (*Figure 2G*).

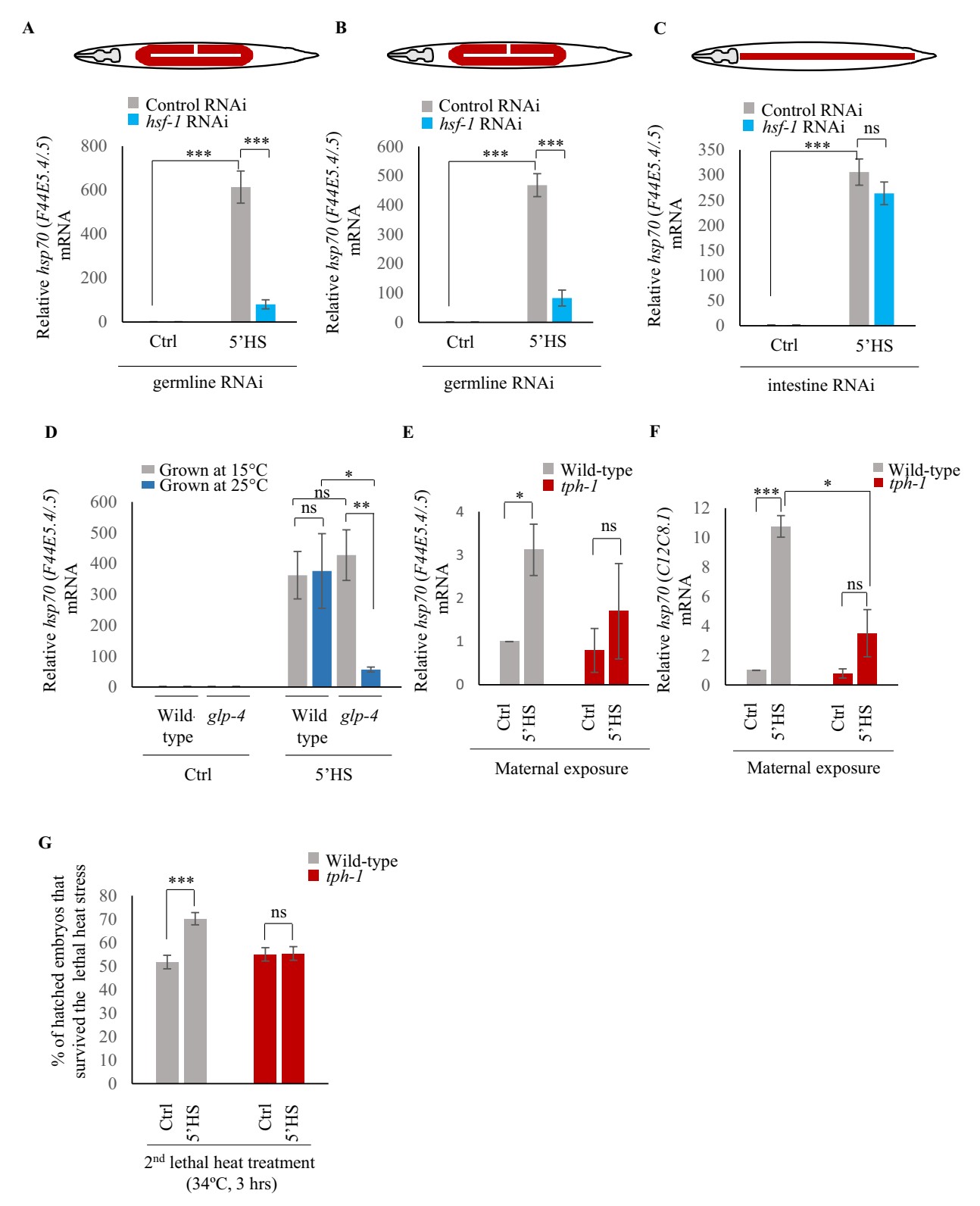

**Figure 2.** Maternal serotonin enables protective gene expression in the germline. (A–C) Average *hsp70* (*F44E5.4/.5*) mRNA levels in control, non heat-shocked (Ctrl) and heat-shocked (5'HS) animals subjected to tissue specific RNAi. mRNA levels were normalized to that in control non-heat shocked, wild-type animals. Heat Shock: 5 min at 34°C. (A) *rrf-1 (pk1714)* animals that predominantly undergo RNAi in germline tissue were subjected to control RNAi and *hsf-1* RNAi (n = 4 experiments). (B) *mkcSi13 [sun-1p::rde-1::sun-1 3'UTR + unc-119(+)] II; rde-1(mkc36) V* animals that predominantly undergo

*Figure 2 continued on next page*

*Figure 2 continued*

RNAi in germline tissue subjected to control RNAi and *hsf-1* RNAi (n = 3 experiments). (C) *rde-1(ne219) V; kbIs7* predominantly undergo RNAi in intestinal tissue subjected to control RNAi and *hsf-1* RNAi (n = 3 experiments). (D) Average *hsp70* (*F44E5.4/.5*) mRNA levels in control non heat-shocked (Ctrl) and heat-shocked (5'HS) wild-type and *glp-4 (bn2) I* animals raised at 15°C (permissive temperature for *glp-4*) or 25°C (restrictive temperature for *glp-4*). n = 4 experiments. mRNA levels were normalized to that in control non-heat shocked animals of same genetic background, raised at the corresponding temperature. Heat Shock: 5 min at 34°C. (E) Average *hsp70* (*F44E5.4/.5*) mRNA levels in embryos laid during a 2 hr interval by wild-type or *tph-1* mutant animals that were not heat shocked (Ctrl), or heat-shocked for 5 min at 34°C and allowed to recover for 2 hr (5'HS). n = 3 experiments, embryos laid from 50 animals/experiment. mRNA levels were normalized to that in control non-heat shocked, wild-type embryos. (F) Average *hsp70* (*C12C8.1*) mRNA levels in embryos laid during a 2 hr interval by wild-type or *tph-1* mutant animals that were not-heat shocked (Ctrl), or heat-shocked for 5 min at 34°C and allowed to recover for 2 hr (5'HS). n = 3 experiments, embryos laid from 50 animals/experiment. mRNA levels were normalized to that in control non-heat shocked, wild-type embryos. (G) Percent larvae from non-heat shocked, or heat-shocked wild-type and *tph-1* mutant animals that survive a subsequent heat exposure to 34°C. Maternal heat shock: 5 min at 34°C. Larval heat shock: 3 hr at 34°C. Note: larval heat exposure was titrated to achieve ~50% lethality amongst progeny of control, non-heat shocked animals. n = 5 experiments; larvae derived from 4 to 5 adult animals/ experiment were scored. Data show Mean ± Standard Error of the Mean. ***, p<0.001; **p<0.01, *p<0.05 (paired Student's t-test). ns, non-significant.

## Serotonin links the stress stimulus to the onset of protective gene expression

Upon the same 5 min exposure to heat, *tph-1* mutant animals differentially expressed only 17, instead of 408 genes as measured by RNA-seq (*Figure 3A*; *Figure 3—figure supplement 1A–C*; *Supplementary file 3*) accumulated less *hsp70* mRNA as measured by qPCR (*Figure 3B,C*) and retained similar transcriptional profiles as unstressed *tph-1* animals by Principal Component Analysis (PCA; *Figure 3—figure supplement 1D*), indicating that they only mildly, if at all, activated HSF-1 in response to the transient temperature change. In contrast to wild-type embryos, embryos from heat-shocked *tph-1* mutant mothers also did not contain more *hsp70* (*F44E5.4/.5*) and *hsp70* (*C12C8.1*) mRNA (*Figure 2E,F*), nor did the larvae display increased stress tolerance (*Figure 2F*). However, although *tph-1* mutant animals were deficient in activating a heat-shock response upon a 5 min exposure to heat stress, they were not deficient in activating HSF-1 *per se*. When exposed to greater intensities of heat stress (15 min exposure to 34°C, instead of 5 min), they accumulated similar levels of *hsp70* mRNA as wild-type animals (*Figure 3D,E*). However, 15 min of heat exposure was already sufficient to impact the viability of germ cells upon fertilization, as only 9 ± 1.5% of the embryos from *tph-1* animals and 33 ± 5% of the embryos from wild type animals (n = 11 experiments, embryos from 4 to 5 animals/experiment) generated during the 2–4 hr time period after mothers had experienced 34°C for 15 min, survived. (*Figure 3F*). This was the case despite the accumulation of equivalent levels of *hsp70* mRNA in both wild-type and *tph-1* mutant mother upon a 15 min heat shock. This suggested that in both wild-type and *tph-1* animals the germ cells were extremely vulnerable to heat-induced damage, and the earlier onset of the protective heat-shock response in germ cells that occurred in wild-type animals upon 5-HT release could maximize germ cell viability.

We have previously shown that in *C. elegans* 5-HT release acts cell-non autonomously to increase *hsp* gene expression. However, how 5-HT released by neurons activates transcription in remote tissues is not known. Therefore to ask how 5-HT may be modulating *hsp* expression in the germline, we used Chromatin immunoprecipitation (ChIP) followed by quantitative PCR (ChIP-qPCR) (*Askjaer et al., 2014*; *Mundade et al., 2014*) to assess the binding of key proteins involved in *hsp* transcription at *hsp* loci, in the presence and absence of 5-HT. Although we conducted ChIP in whole animals, we leveraged the fact that the majority of *hsp* transcription during 5 min of heat-shock occurred in the germ cells to infer that any changes in ChIP occupancy at *hsp* genes, if not reporting exclusively on what occurred in germline chromatin, would at the very least, be representative of changes at *hsp* loci in the germline.

The differences in the onset of transcription between wild-type animals and *tph-1* mutant animals was reflected by differences in the occupancy of RNA polymerase II (RNAP) and HSF-1 at *hsp70* (*F44E5.4/.5*) and *hsp70* (*C12C8.1*), as assessed by ChIP-qPCR (*Askjaer et al., 2014*; *Mundade et al., 2014*) using primer sets targeted to the Promoter region of these *hsp* genes and, for RNAP, also to two regions within the gene body (*Figure 4*; *Figure 4—figure supplement 1A*). In contrast to *Drosophila* and mammalian cells (*Adelman and Lis, 2012*; *Spencer and Groudine, 1990*), in *C. elegans*, RNAP pausing is rare in the absence of stress such as starvation (*Kruesi et al., 2013*; *Maxwell et al.,*

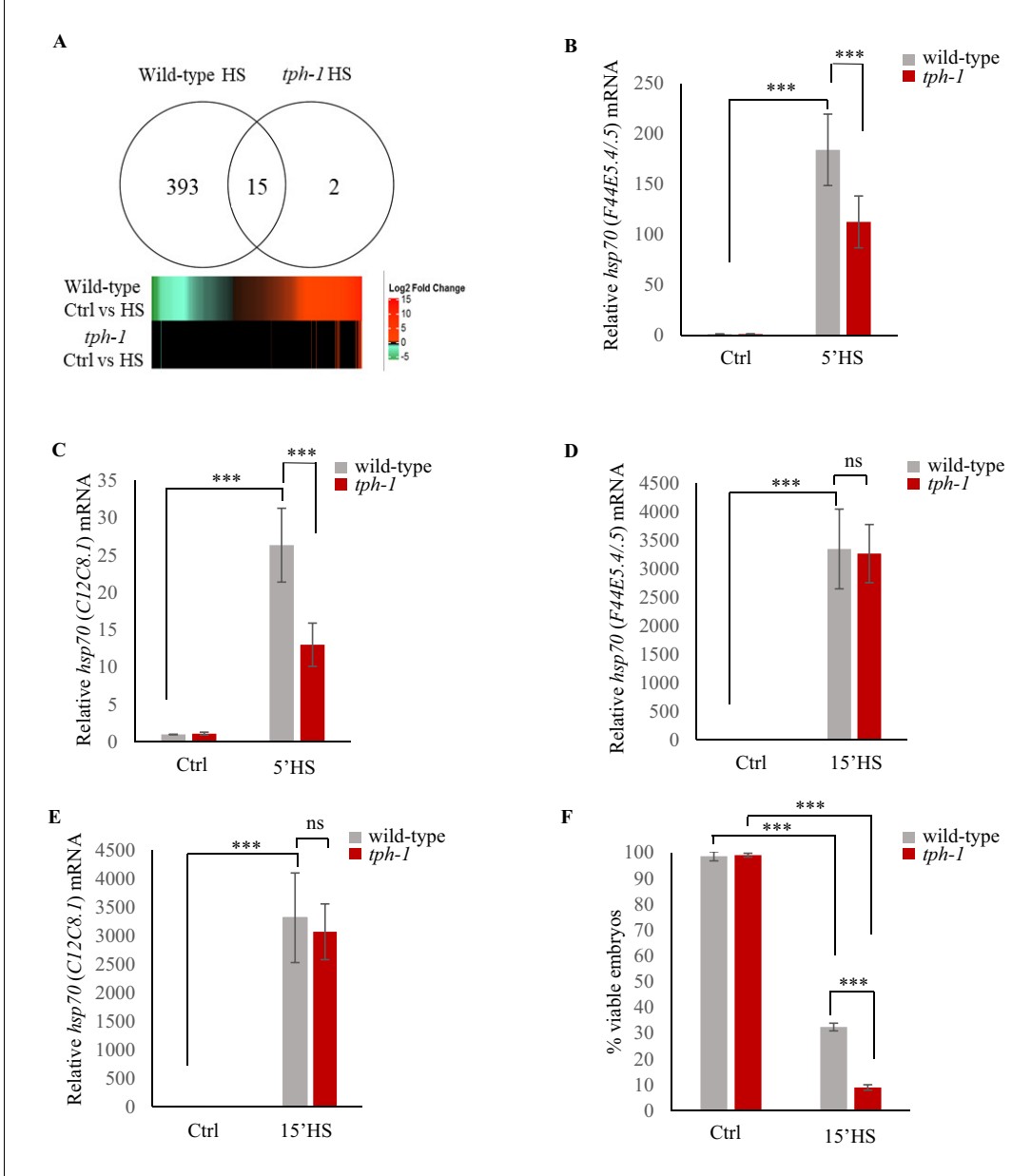

**Figure 3.** Serotonin accelerates the onset of gene expression upon heat shock. (A) Top: Venn diagram showing overlap between genes differentially expressed in wild-type animals and *tph-1* mutants (0.01 FDR) following 5 min heat-shock at 34˚C. Numbers depict differentially induced genes in each strain. Data from RNA-seq experiments. Bottom: Heat map depicting expression levels (Log2 Fold change) of differentially expressed genes in wild-type and *tph-1* mutants. (B–E) Average *hsp70* mRNA in wild-type and *tph-1* mutant animals following heat shock: (B) *hsp70* (*F44E5.4/.5*) mRNA and (C) *hsp70* (*C12C8.1*) mRNA levels in wild-type and *tph-1* mutant animals under control conditions and following heat shock at 34˚C for 5 min (n = 22 experiments). (D) *hsp70* (*F44E5.4/.5*) mRNA and (E) *hsp70* (*C12C8.1*) mRNA levels in wild-type and *tph-1* mutant animals under control conditions and following heat shock at 34˚C for 15 min (n = 5 experiments). (B–E) mRNA levels were normalized to that in control non-heat shocked, wild-type animals. (F) Percent viable embryos from control (Ctrl) and heat-shocked (15'HS), wild-type animals and *tph-1* mutant animals. Embryos were laid during a 2 hr interval by non-heat shocked animals, or animals that were heat-shocked for 15 min at 34˚C and allowed to recover at 20˚C for 2 hr. n = 4 experiments, 4–5 animals/experiment. Data in B–F show Mean ± Standard Error of the Mean. ***, p<0.001 (paired Student's t-test). ns, non-significant. The online version of this article includes the following figure supplement(s) for figure 3:

**Figure supplement 1.** The lack of serotonin diminishes HSF-1-dependent gene expression upon 5 min heat-shock.

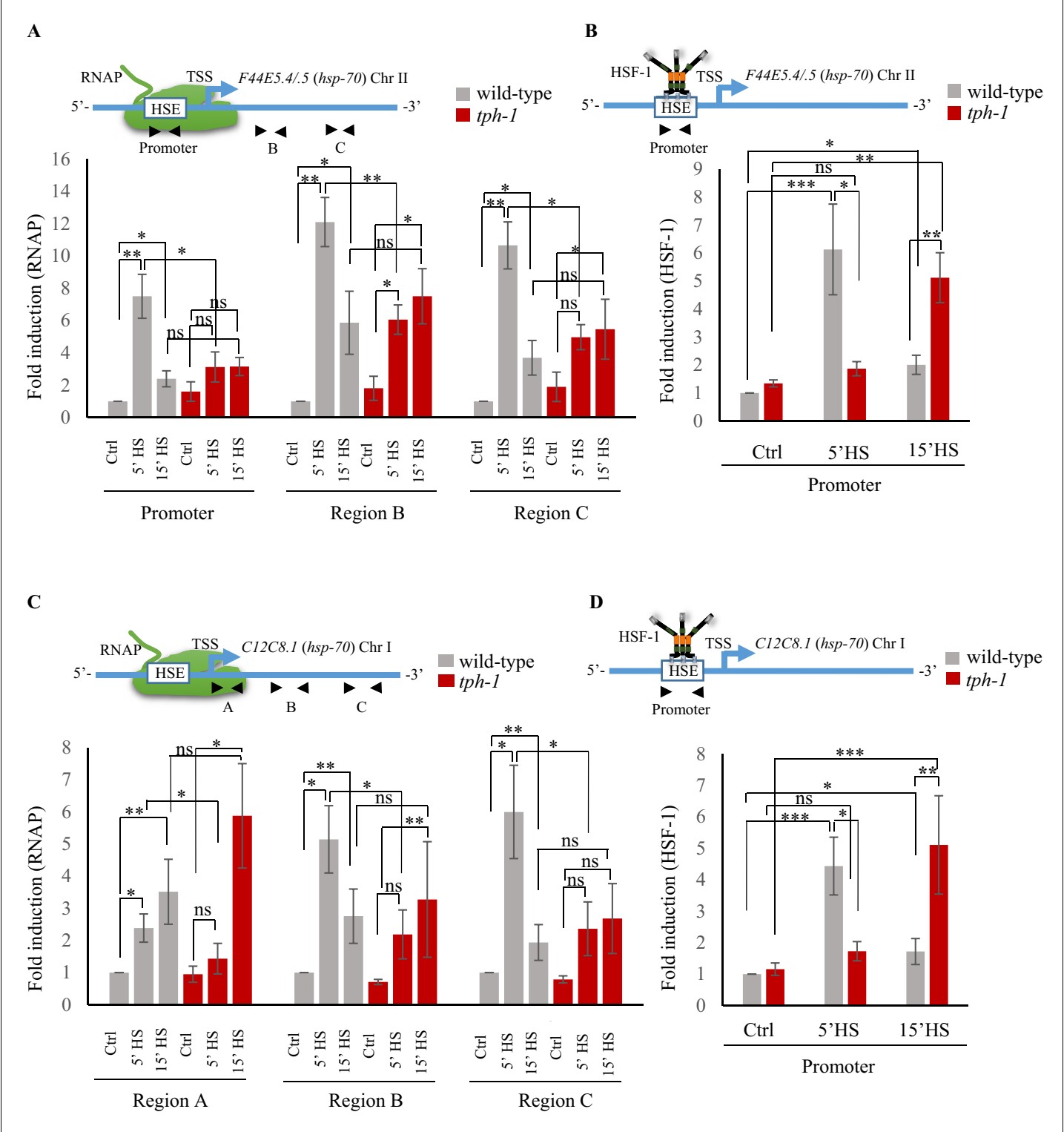

**Figure 4.** Serotonin accelerates the onset of RNAP and HSF-1 recruitment to target genes. (**A**) Top: Schematic of *hsp70* (*F44E5.4/.5*) gene regions within the Promoter (−390 to −241), middle of gene (Region B: +696 to +915) and toward 3'-UTR (Region C: +1827 to +1996) that were assayed for occupancy by RNAP. Bottom: RNAP occupancy at Promoter, Region B and Region C in wild-type animals and *tph-1* mutants following exposure to 34°C for 5 min and 15 min (n = 8 experiments). (**B**) Top: Schematic of *hsp70* (*F44E5.4/.5*) gene regions within the Promoter (−390 to −241) assayed for occupancy of HSF-1. This is the same Promoter region as in A. Bottom: HSF-1 occupancy at the Promoter in wild-type animals and *tph-1* mutants following exposure to 34°C for 5 min and 15 min (n = 14 experiments). (**C**) Top: Schematic of *hsp70* (*C12C8.1*) gene regions close to the beginning (Region A: +25 to +185), middle of gene (Region B: +475 to +583) and towards 3'-UTR (Region C:+1645 to +1835) assayed for occupancy by RNAP.
*Figure 4 continued on next page*

*Figure 4 continued*

Bottom: RNAP occupancy at Region A, Region B and Region C in wild-type animals and *tph-1* mutants following exposure to 34°C for 5 min and 15 min (n = 8 experiments). (D) Top: Schematic of *hsp70* (*C12C8.1*) gene region within the Promoter (−166 to −78) assayed for HSF-1 occupancy. Bottom: HSF-1 occupancy at the Promoter in wild-type animals and *tph-1* mutants following exposure to 34°C for 5 min and 15 min (n = 14 experiments). Data show Mean ± Standard Error of the Mean. Data in all experiments are normalized to values from control (non-heat shocked) wild-type animals. Specificity and efficiency of pull-down under control conditions was ascertained (see *Figure 4—figure supplement 2A, (B*). *, p<0.05; **, p<0.01 ***, p<0.001; (ANOVA with Tukey's correction). ns, non-significant.

The online version of this article includes the following figure supplement(s) for figure 4:

**Figure supplement 1.** Characterization of RNAP and HSF-1 for ChIP-qPCR assays.

**Figure supplement 2.** Validation of specificity and efficiency of antibodies used for ChIP assays.

*2014*), and that was evident in the even distribution of RNAP across three distinct regions ('Region A/Promoter, Region B and Region C) of the *hsp70* genes under basal, non-heat shock conditions (*Figure 4—figure supplement 1A*). In wild-type animals, consistent with the rapid induction of mRNA, RNAP was recruited to *hsp70* (*F44E5.4/.5*) and *hsp70* (*C12C8.1*), peaking within 5 min of exposure to heat and remains significantly enriched at these genes upon continued heat exposure (*Figure 4A,C*). The exact pattern of enrichment differed between the two *hsp70* genes, for reasons that are unclear. Notwithstanding these differences, in *tph-1* mutants, RNAP occupancy was significantly lower than that in wild-type animals upon 5 min of heat shock (*Figure 4A,C*), but accumulated to similar or even higher levels as in wild-type animals upon continued heat exposure (15 min exposure to 34°C; *Figure 4A,C*). Similarly, in wild-type animals HSF-1 was enriched at the promoter regions of *hsp* genes by 5 min of exposure of animals to heat, and remained enriched, although to lesser amounts by 15 min. In *tph-1* mutant animals HSF-1 was not recruited to these *hsp* gene promoters by 5 min, and HSF-1 levels at *hsp* promoters of *tph-1* mutants exposed to a 5 min heat-shock did not significantly differ from that in control non-heat shocked *tph-1* animals (*Figure 4B,D*). However, by 15 min following heat-shock, HSF-1 enrichment at *hsp* promoters in *tph-1* mutant animals was similar to that in wild-type animals after a 5 min heat shock, suggesting that the binding of HSF-1 to its promoter was delayed in the absence of 5-HT (*Figure 4B,D*). The latter was evaluated by ChIP-qPCR using animals that expressed endogenous HSF-1 tagged at the C-terminus with 3X FLAG (*Figure 4—figure supplement 1B*). The enrichment of HSF-1 at target genes was specific and not apparent at the *syp-1* promoter that did not contain an HSF-1 binding site (*Figure 4—figure supplement 1C*), and was not a consequence of differences in HSF-1 protein levels between wild-type and *tph-1* mutants (*Figure 4—figure supplement 1D,E*).

These data, together, supported a model whereby the release of maternal 5-HT in wild-type animals triggered an earlier onset of transcription. This difference in timing of the onset of transcription was reflected by differences in the timing of HSF-1 and RNAP occupancy at *hsp* genes: in wild-type animals a 5 min exposure was sufficient to induce a robust occupancy of HSF-1 protein and RNAP, while in the absence of 5-HT, binding of both HSF-1 and RNAP did occur, but were delayed. Taken together with the observation that the 5 min heat-shock activated HSF-1-dependent gene expression predominantly in germ cells of wild-type animals, these data suggested that 5-HT was responsible for the timing of HSF-1 activation in germ cells.

## Serotonin-dependent recruitment of FACT to displace histones hastens the onset of transcription

One mechanism by which 5-HT might accelerate the onset of transcription would be to alter chromatin accessibility to allow the transcription factor and RNAP to bind chromatin (*Fujimoto et al., 2012*; *Lis and Wu, 1993*; *Roberts et al., 1987*; *Teves and Henikoff, 2013*; *Wang et al., 2018*; *Zhao et al., 2005*). To test if this was the case, we conducted ChIP-qPCR to examine levels of the histone H3, a component of the core nucleosome, at the two *hsp70* genes in wild-type animals and *tph-1* mutants upon transient exposure to heat (*Figure 5A*; *Figure 5—figure supplement 1A*). In the absence of heat shock, H3 levels at the promoter, Transcription Start Site (TSS) and gene body (Region B) of *hsp70* genes (*Zhao et al., 2005*) were comparable between wild-type animals and *tph-1* mutants, although *tph-1* mutant animals have slightly higher, but not significantly higher, levels throughout. In wild-type animals, the brief exposure to heat disrupted histone-DNA interactions across the entire *hsp70* genes: H3 occupancy at the promoter, TSS and gene body decreased

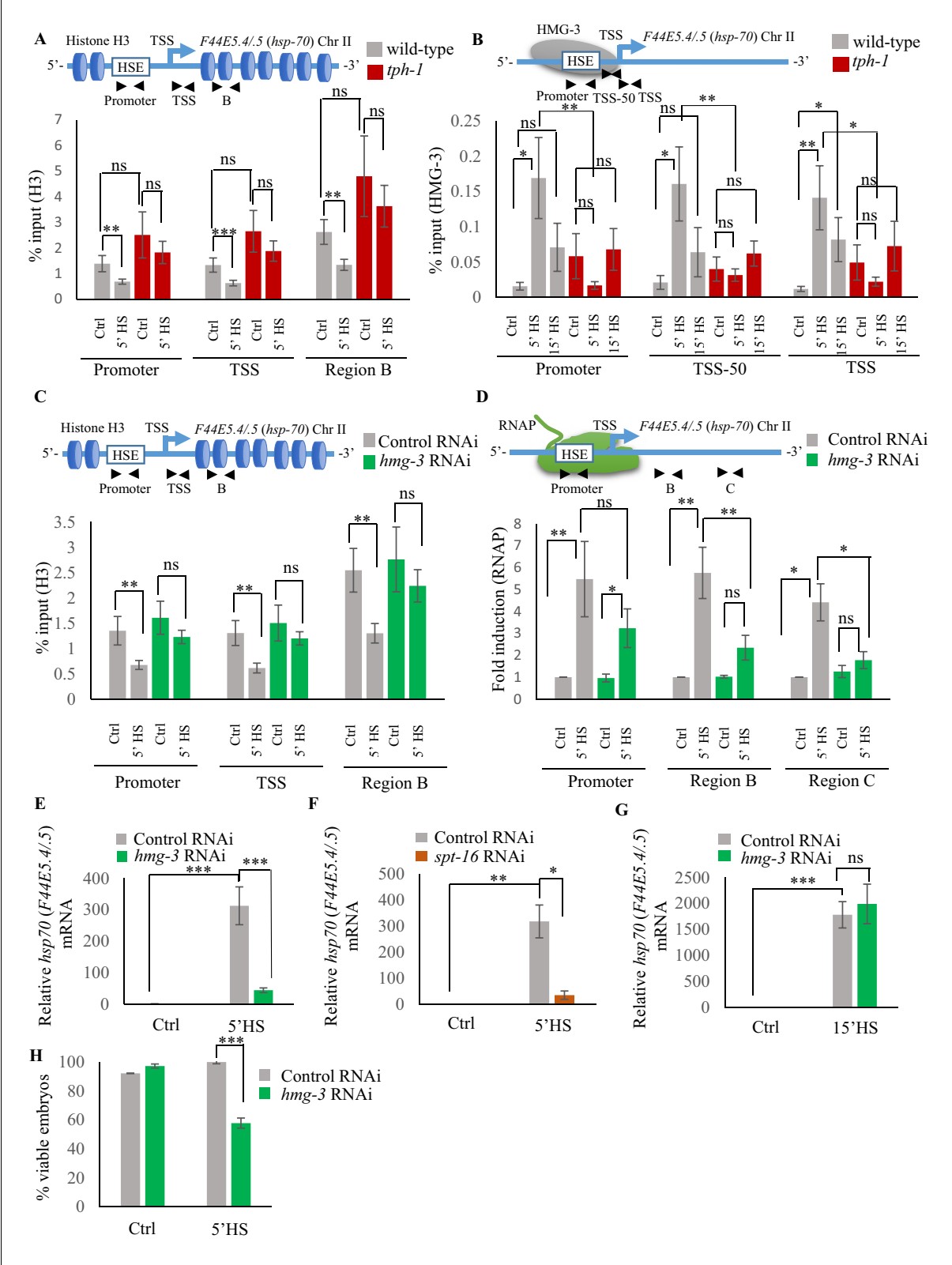

**Figure 5.** Serotonin enables FACT recruitment at *hsp* genes to displace nucleosomes and hasten the onset of transcription. (**A**) Top: Schematic of *hsp70* (*F44E5.4/.5*) gene regions within the Promoter (same as *Figure 4A and B*: −390 to −241), Transcription Start Site (TSS: −81 to +38) and gene body (same as Region B in *Figure 4A*: +696 to +915) assayed for histone H3 occupancy. Bottom: Occupancy of histone H3 (% input) at Promoter, TSS and Region B upon 5 min at 34°C (n = 9 experiments). (**B**) Top: Schematic of *hsp70* (*F44E5.4/.5*) gene: regions in the Promoter (same as in *Figure 4B*:
*Figure 5 continued on next page*

Figure 5 continued

−390 to −241), region upstream of Transcription Start Site (TSS-50: −221 to −63) and Transcription Start Site (Same as **A**): TSS: −81 to +38) were assayed for HMG-3 occupancy. Bottom: HMG-3 occupancy (% input) across Promoter, TSS-50 and TSS following 5 and 15 min at 34°C (n = 13 experiments). (**C**) Top: Schematic of *hsp70* (*F44E5.4/.5*) gene regions assayed for H3 occupancy (same as in **A**). Bottom: Occupancy of histone H3 in Promoter, TSS and in Region B of *hsp70* (*F44E5.4/.5*) under control conditions and following 5 min at 34°C, in control-RNAi treated animals and *hmg-3*-RNAi treated animals (n = 9 experiments). (**D**) Top: Schematic of *hsp70* (*F44E5.4/.5*) gene: same regions as in **Figure 4A**, that is Promoter (−390 to −241), Region B (+696 to +915) and Region C (+1827 to +1996) were assessed for RNAP occupancy in control-RNAi treated and *hmg-3*-RNAi treated animals. Bottom: Fold change in RNAP across Promoter, Region B and Region C following 5 min heat shock at 34°C (n = 5 experiments). % input values were normalized to that in control-RNAi treated animals at Promoter and Regions B and C. Specificity and efficiency of pull-down under control conditions were verified (see **Figure 5—figure supplement 2A**, (**B**). (**E**) *hsp70* (*F44E5.4/.5*) mRNA levels in control-RNAi treated and *hmg-3* -RNAi treated animals following a 5-min heat shock at 34°C (n = 6 experiments). (**F**) *hsp70* (*F44E5.4/.5*) mRNA levels in control-RNAi treated and *spt-16* -RNAi treated animals following a 5 min heat shock at 34°C (n = 4 experiments). (**G**) *hsp70* (*F44E5.4/.5*) mRNA levels in control-RNAi treated and *hmg-3* -RNAi treated animals following a 15 min heat shock at 34°C (n = 6 experiments) (**H**) Percent viable embryos (laid 2–4 hr post-5 min heat shock at 34°C) from control-RNAi treated and *hmg-3*-RNAi treated animals (n = 5 experiments, 4–5 animals/experiment). Data show Mean ± Standard Error of the Mean. *, p<0.05; **, p<0.01 ***, p<0.001; (**A–D**, ANOVA with Tukey's correction; **E–H**) Paired Student's t-test). ns, non-significant.

The online version of this article includes the following figure supplement(s) for figure 5:

**Figure supplement 1.** Characterizing H3 and FACT occupancy across *hsp* genes in wild-type and *tph-1* mutant animals under control and heat shock conditions.

**Figure supplement 2.** Validation of specificity and efficiency of antibodies used for ChIP assays.

significantly upon the 5 min heat exposure as would be required to allow HSF-1 and RNAP access to DNA (**Figure 5A**; **Figure 5—figure supplement 1A**). In contrast, in *tph-1* mutant animals that lack 5-HT, H3 occupancy did not decrease upon heat shock but instead remained similar to that under basal, non-heat-shock conditions (**Figure 5A**; **Figure 5—figure supplement 1A**). This suggested that changes in chromatin accessibility could underlie the differences in the response times of wild-type animals and *tph-1* mutants.

The histone chaperone FACT (**Adelman et al., 2006**; **Belotserkovskaya et al., 2003**; **Belotserkovskaya et al., 2004**; **Buckley and Lis, 2014**; **Formosa, 2012**; **Fujimoto et al., 2012**; **Kishimoto et al., 2017**; **Smith et al., 2004**; **Svejstrup, 2003**; **Van Lijsebettens and Grasser, 2010**), a complex of two proteins, SPT16 and SSRP1, is known to disassemble histones to facilitate RNAP transcription at stress genes. In mammalian cells, FACT associates with HSF1 through its interaction with RPA (Replication Protein A), to allow the transcription factor access to DNA at the promoter (**Fujimoto et al., 2012**). In *C. elegans*, the SSRP1 subunit of FACT consists of HMG-3 and HMG-4, of which HMG-3 is expressed exclusively in the germline, and HMG-4 in somatic tissue. HMG-3/HMG-4 along with SPT16, which is ubiquitously expressed, have been shown to displace nucleosomes and epigenetically modulate gene expression (**Kolundzic et al., 2018**; **Suggs et al., 2018**). To investigate whether the observed difference in H3 loss between wild-type animals and *tph-1* mutants was mediated by FACT activity in germ cells, we examined HMG-3 occupancy at *hsp* genes in wild-type animals and *tph-1* mutant animals in strains expressing HMG-3 tagged at its endogenous locus with a 3X hemagglutinin (HA) tag (**Kolundzic et al., 2018**; **Figure 5B**; **Figure 5—figure supplement 1B**). Since HMG-3 is expressed exclusively in the germline of *C. elegans*, these data allowed us to make specific conclusions about the effects of 5-HT on germ cell chromatin.

As with HSF-1, HMG-3 protein levels are similar in wild-type and *tph-1* mutant animals (**Figure 5—figure supplement 1C,D**). Nevertheless in wild-type animals, but not in *tph-1* mutants, HMG-3 was recruited to *hsp* genes by 5 min of heat shock (**Figure 5B**; **Figure 5—figure supplement 1B**), and was necessary for the displacement of H3 histones at the Promoter, TSS and gene body as seen upon decreasing HMG-3 levels using RNAi (**Figure 5C**; **Figure 5—figure supplement 1E**). HMG-3 was also necessary for RNAP occupancy at *hsp* genes upon 5 min of heat-exposure as RNAi-induced down-regulation of *hmg-3* decreased RNAP occupancy across most regions of these genes. (**Figure 5D**; **Figure 5—figure supplement 1F**). RNAP at Region A of *hsp70* (*C12C8.1*) was, for unknown reasons, not affected by *hmg-3* knock-down. Notwithstanding, the expression levels of *hsp* genes was diminished upon *hmg-3* RNAi (**Figure 5E**; **Figure 5—figure supplement 1G**). A similar decrease in *hsp* gene expression upon the 5 min heat shock was seen upon decreasing the levels of the HMG-3 interacting partner SPT-16 (**Figure 5F**; **Figure 5—figure supplement 1H**), suggesting

that HMG-3 and SPT-16 acted as a complex (FACT) to promote HSF-1-dependent gene expression in the germline.

In *tph-1* mutant animals HMG-3 was not recruited to *hsp* genes at significant levels either after 5 or 15 min of heat-shock (*Figure 5B*; *Figure 5—figure supplement 1B*) suggesting that gene expression that occurred in *tph-1* mutants upon continued heat stress (*Figure 3D,E*) likely occurred through a HMG-3-independent mechanism. This was also supported by the observation that RNAi-induced downregulation of *hmg-3* levels in wild-type animals impaired *hsp* mRNA induction upon 5 min of heat shock (*Figure 5E*; *Figure 5—figure supplement 1G*) but did not significantly affect *hsp* mRNA accumulation after 15 min (*Figure 5G*; *Figure 5—figure supplement 1I*). Once again, even though HMG-3 was only required for the early onset of HSF-1 activation, and not for its activation per se, HMG-3 was required to protect germ cells from transient temperature fluctuations, as decreasing HMG-3 levels using RNAi decreased progeny survival upon transient heat shock much the same way as the lack of 5-HT or HSF-1 (*Figure 5H*).

The role of 5-HT in HMG-3 recruitment was confirmed by experiments where *hsp* gene expression (*Ooi and Prahlad, 2017*; *Tatum et al., 2015*) was induced by optogenetically activating the ADF and NSM serotonergic neurons to release 5-HT (*Figure 6—figure supplement 1A–C*). RNAi induced knock-down of *hmg-3* levels dramatically abrogated the 5-HT-dependent increase in *hsp* mRNA (*Figure 6A*; *Figure 6—figure supplement 1D*). In mammalian cells, FACT is targeted to *hsp* promoters through its indirect interaction with HSF1 *via* RPA (*Fujimoto et al., 2012*). In *C. elegans* also FACT recruitment to *hsp* genes depended directly or indirectly on HSF-1 as RNAi-dependent downregulation of *hsf-1* decreased FACT recruitment at *hsp* genes (*Figure 6B*; *Figure 6—figure supplement 1E*). These data together indicated that 5-HT-signaling enabled HSF-1 to recruit HMG-3 in germ cells and displace histones to shorten the onset of RNAP-dependent gene expression and ensure viability of germ cells during stress.

## Serotonin-induced PKA-activation is a conserved signaling pathway that enables HSF1 to recruit FACT

To identify the intracellular signal transduction pathway triggered by 5-HT to enable the interaction of HSF-1 with FACT, we decided to use mammalian cells (*Figure 7—figure supplement 1A*) where we would be able to isolate cell autonomous effects away from cell non-autonomous effects, and leverage the wealth of information about mammalian HSF1 (*Anckar and Sistonen, 2011*; *Batulan et al., 2003*; *Budzyński et al., 2015*; *Cotto et al., 1996*; *Holmberg et al., 2002*; *Jurivich et al., 1994*; *Li et al., 2017*; *Zheng et al., 2016*). As in *C. elegans*, exposure of mammalian cells to exogenous 5-HT could also autonomously activate HSF1. Treatment of mouse primary cortical neurons (*Garrett et al., 2012*; *Garrett and Weiner, 2009*; *Molumby et al., 2016*) with exogenous 5-HT resulted in a dose- and time-dependent increase in mRNA levels of the most highly inducible *hsp* genes that are targets of mammalian HSF1: *Hspb1* (*Figure 7—figure supplement 1B*), *Hspa1a* (*Figure 7A*) and *Hspb5* (*Figure 7—figure supplement 1C*). A similar increase in *HSPA1A* mRNA (*Figure 7B*) and HSPA1A protein levels (*Figure 7—figure supplement 1D,E*) was observed upon treatment of human NTera2 (NT2) cells. siRNA induced knock-down of HSF1 (*Figure 7—figure supplement 1F*) abrogated 5-HT-induced *HSPA1A* mRNA expression (*Figure 7C*). Thus, remarkably, acute increases in 5-HT activated HSF1-dependent gene expression in mammalian cells, much the same way it did in *C. elegans*.

The effects of 5-HT are transduced through intracellular signal transduction pathways and depend on the particular 5-HT receptor(s) involved in the biological process (either G protein-coupled receptors—GPCRs, or ligand-gated ion channels) (*Julius, 1991*; *Tecott and Julius, 1993*). Therefore, to identify the intracellular pathway involved in 5-HT-induced HSF1 activation, we used a panel of 5-HT receptor specific agonists (*Bardin, 2011*; *Bauer, 2015*; *Cortes-Altamirano et al., 2018*; *Iwanir et al., 2016*; *Meneses and Hong, 1997*; *Nikiforuk, 2015*; *Tecott and Julius, 1993*; *Tierney, 2001*). Agonists of 5-HT4 receptor (BIMU8), but not 5-HT6, 5-HT2A and 5-HT1 elicited a dose-(*Figure 7D*) and time-(*Figure 7—figure supplement 1G*) dependent increase in *HSPA1A* mRNA in NT2 cells that was HSF-1 dependent (*Figure 7—figure supplement 1H*), mimicking the effects of 5-HT. BIMU8 also induced *Hspa1a* and *Hspb1* mRNA in primary cortical neurons (*Figure 7—figure supplement 1I*). The 5-HT4 receptor is a GPCR that signals though adenylyl cyclase and activates protein kinase A (PKA) (*Bockaert et al., 2011*; *Cortes-Altamirano et al., 2018*; *Lalut et al., 2017*; *Lucas, 2009*; *Meneses, 2013*; *Meneses and Hong, 1997*). PKA has been shown to phosphorylate

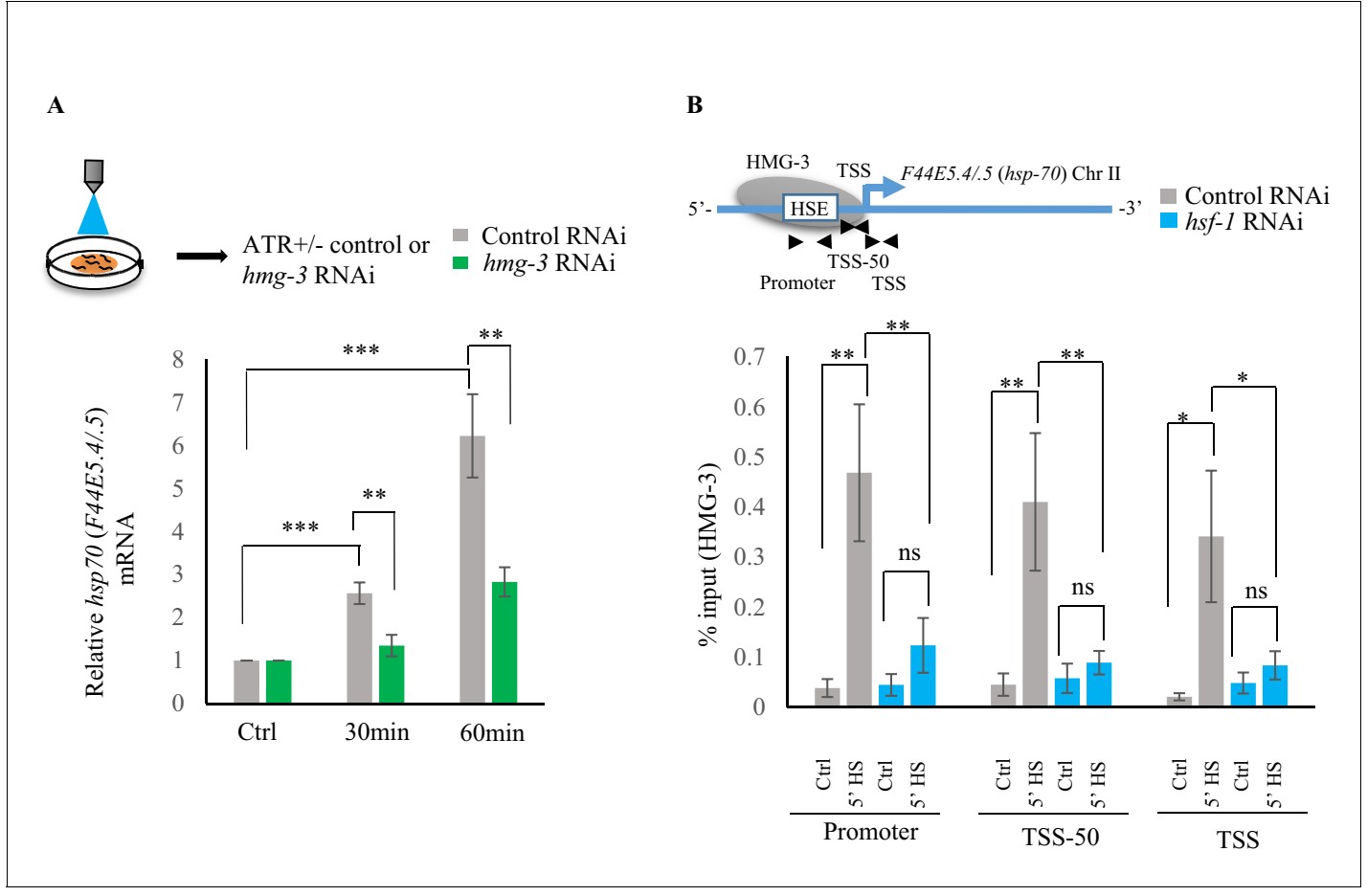

**Figure 6.** Serotonin release mediates FACT recruitment by HSF-1 to induce *hsp* expression. (**A**) Top: Schematic of optogenetic activation of 5-HT release conducted by stimulating ADF and NSM neurons in control-RNAi treated and *hmg-3*-RNAi treated animals. Bottom: *hsp70* (*F44E5.4/.5*) mRNA levels in control-RNAi treated and *hmg-3*-RNAi treated animals at different time points following optogenetic stimulation. mRNA levels were normalized to control-RNAi treated and *hmg-3*-RNAi treated unstimulated animals respectively (n = 6 experiments). (**B**) Top: Schematic of *hsp70* (*F44E5.4/.5*) gene Promoter, TSS-50 and TSS to assess HMG-3 occupancy in control-RNAi treated and *hsf-1*-RNAi treated animals. Bottom: HMG-3 occupancy (% input) at Promoter, TSS-50 and TSS in control-RNAi and *hsf-1*-RNAi treated animals following 5 min at 34˚C (n = 9 experiments). Specificity and efficiency of pull-down under control conditions was ascertained. Data show Mean ± Standard Error of the Mean. *, p<0.05; **, p<0.01 ***, p<0.001. (**A**) Paired Student's t-test. (**B**) ANOVA with Tukey's correction). ns, non-significant.
The online version of this article includes the following figure supplement(s) for figure 6:

**Figure supplement 1.** Serotonin-induced transcriptional activity of HSF-1 in *C. elegans* is FACT dependent.

mammalian HSF1 at the serine 320 residue during heat shock (*Melling et al., 2006*; *Murshid et al., 2010*; *Zhang et al., 2011*). In agreement with this, BIMU8 treatment triggered an increase in S320-modified HSF1 (*Murshid et al., 2010*; *Zhang et al., 2011*) as detected by a phospho-specific antibody (*Figure 7E,F*). Inhibiting PKA activity using the drug H89 (*Lochner and Moolman, 2006*) inhibited the BIMU8-induced increase in S320 phosphorylation (*Figure 7E,F*). In addition, BIMU8 treatment promoted the nuclear localization of HSF1 which in turn could also be inhibited by H89, recapitulating previous studies on PKA-induced phosphorylation of HSF1 (*Murshid et al., 2010*; *Figure 7G*; *Figure 7—figure supplement 1J*). *HSPA1A* mRNA levels that were induced by BIMU8 treatment were also inhibited upon treatment of the cells with H89 (*Figure 7H*). Moreover, as with 5-HT induced activation of HSF-1 in *C. elegans,* BIMU8-induced activation of HSF1 in NT2 cells also required FACT, and the knockdown of the *SUPT16H* subunit of FACT by siRNA (*Figure 7— figure supplement 1K*) abrogated the BIMU8-induced upregulation of *HSPA1A* mRNA (*Figure 7I*). These data together indicate that 5-HT cell-autonomously enables HSF1 to recruit FACT in mammalian cells through the activation of 5-HT4 receptor and the conserved cAMP-PKA intracellular

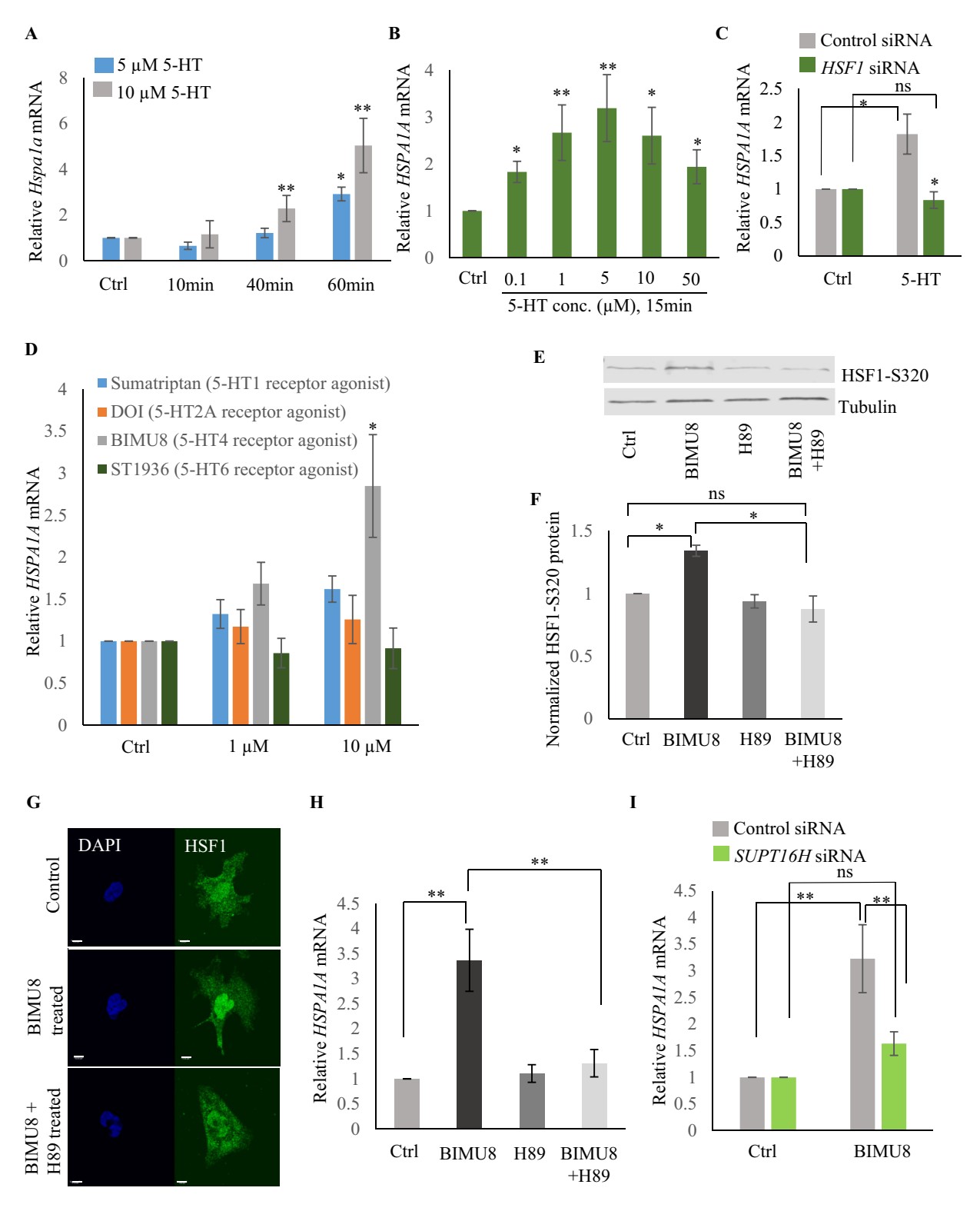

**Figure 7.** Serotonin activates PKA-mediated signal transduction to enable HSF1-FACT interaction in mammalian cells. (A) Time and dose-dependent change in *Hspa1a* mRNA levels in control and 5-HT treated primary cortical neuronal cultures (n = 4 experiments). (B) Dose-dependent change in *HSPA1A* mRNA levels in control NT2 cells and NT2 cells treated with 5-HT for 15 min (n = 4 experiments). (C) *HSPA1A* mRNA levels in NT2 cells treated with 5 μM 5-HT for 15 min, transfected with control and *HSF1* siRNA (n = 4 experiments). (D) *HSPA1A* mRNA levels in NT2 cells treated with two

*Figure 7 continued on next page*

*Figure 7 continued*

different doses of four 5-HT receptor agonists relative to control untreated cells (n = 5 experiments). NT2 cells were treated for 10 min. (**E–F**) Protein levels of S320 phospho-modified HSF1 in control NT2 cells and cells treated with 10 µM BIMU8 for 10 min, in the presence or absence of the PKA inhibitor, H89 (n = 4 experiments). (**E**) Representative western blot using an antibody that recognizes HSF1 phosphorylated at S320. Tubulin served as the internal control. (**F**) Quantitation of phospho-S320 levels (n = 4 experiments). (**G**) Representative micrographs showing projections of confocal images of HSF1 localization in control NT2 cells and cells treated with 10 µM BIMU8 for 10 min, in the presence or absence of the H89 (n = 2 experiments; 25 cells). Scale bar = 10 µm. (**H**) *HSPA1A* mRNA levels relative to control NT2 cells upon treatment with 10 µM BIMU8 for 10 min, in the presence or absence of H89 (n = 5 experiments). (**I**) *HSPA1A* mRNA levels in cells treated with 10 µM BIMU8 for 10 min, transfected with control and *SUPT16H* siRNA. mRNA levels and protein levels are normalized to control RNAi-treated or unstimulated cells (n = 5 experiments). Data in A-D, F, H, I show Mean ± Standard Error of the Mean. *, p<0.05; **, p<0.01 ***, p<0.001; (Paired Student's t-test). ns, non-significant.

The online version of this article includes the following figure supplement(s) for figure 7:

**Figure supplement 1.** Serotonin activates a PKA-mediated signal transduction pathway to enable HSF1-FACT interaction in mammalian cells.

---

signaling pathway, and as in *C. elegans* this allowed HSF1 to access *hsp* genes and initiate RNAP-dependent gene expression even in the absence of stress.

Although *C. elegans* do not possess a 5-HT4 receptor ortholog, they possesses 5-HT receptors that are distributed throughout somatic tissue, and can activate PKA (*Chase and Koelle, 2007*). Therefore, to examine whether in *C. elegans* also, 5-HT acted through PKA to accelerate the onset of HSF-1-dependent gene expression we modulated the *C. elegans* PKA holoenzyme. PKA exists as a tetramer with catalytic and regulatory subunits, and the release of inhibition by the regulatory subunits results in the enabling of the catalytic activity of PKA. In *C. elegans* *kin-1* encodes the catalytic subunits of PKA, and inhibition of *kin-1* diminishes PKA activity, while *kin-2* encodes the regulatory subunits, and decreasing *kin-2* levels releases KIN-1 and activates PKA (*Gottschling et al., 2017*; *Liu et al., 2017*; *Steuer Costa et al., 2017*; *Wang and Sieburth, 2013*; *Xiao et al., 2017*; *Zhou et al., 2007*). Decreasing *kin-1* levels by RNAi dampened the induction of *hsp70* mRNA that occurs upon optogenetic activation of 5-HT release (*Figure 8A*; *Figure 8—figure supplement 1A*). Decreasing *kin-1* levels by RNAi also prevented the recruitment of HMG-3 in germ cells by HSF-1 after transient exposure to heat (*Figure 8B*; *Figure 8—figure supplement 1B*). Moreover, as in mammalian cells the role of KIN-1 in activating HSF-1 appeared to be cell autonomous, as decreasing *kin-1* levels only in germ cells decreased *hsp70* mRNA levels upon 5 min heat-shock, similar to decreasing *kin-1* levels in whole animals (*Figure 8C,D*; *Figure 8—figure supplement 1C,D*). Conversely, activating PKA in *tph-1* animals by knocking down *kin-2* rescued the delayed response of *tph-1* mutant animals, both increasing occupancy of HSF-1 at *hsp* promoters by 5 min upon heat exposure despite the absence of 5-HT (*Figure 8E*; *Figure 8—figure supplement 1E*), and increasing *hsp70* (F44E5.4/.5) mRNA levels to wild-type levels upon 5 min heat shock (*Figure 8F*; *Figure 8—figure supplement 1F*). Activating PKA by RNAi-mediated knockdown of *kin-2* also rescued, significantly albeit incompletely, the embryonic lethality induced by exposing *tph-1* mutant animals to 5 min of heat (*Figure 8—figure supplement 1G*).

These data together allow us to propose a model whereby maternal 5-HT released by neurons acts on germ cell through 5-HT-mediated PKA signaling to hasten the timing of stress-gene expression (*Figure 7G*). This occurs by the PKA-dependent phosphorylation of HSF-1 which enables it to recruit FACT, displace nucleosomes and promote RNAP transcription though chromatin. The activation of HSF-1 by PKA signaling occurs in the germline, as knocking down *kin-1* only in germ cells is enough to compromise *hsp* mRNA expression upon the 5 min heat-shock. Although we do not show it, we hypothesize that 5-HT likely acts directly on the germline cells upon release from neurons due to its ability to diffuse through the coelomic fluid and bind 5-HT receptors in the germline. However, it is also possible that 5-HT binds to receptor on other tissues, which in turn secrete signals to activate PKA in the germline and accelerate HSF1-dependent transcription. In either case, this leads to enhanced thermotolerance of the progeny of heat-shocked mothers (*Figure 7G*).

## Discussion

One of the more recent developments in the regulation of stress responses has been the demonstration that in *C. elegans* the activation of the unfolded protein response (UPR) in the cytoplasm (*Ooi and Prahlad, 2017*; *Prahlad et al., 2008*; *Tatum et al., 2015*), endoplasmic reticulum

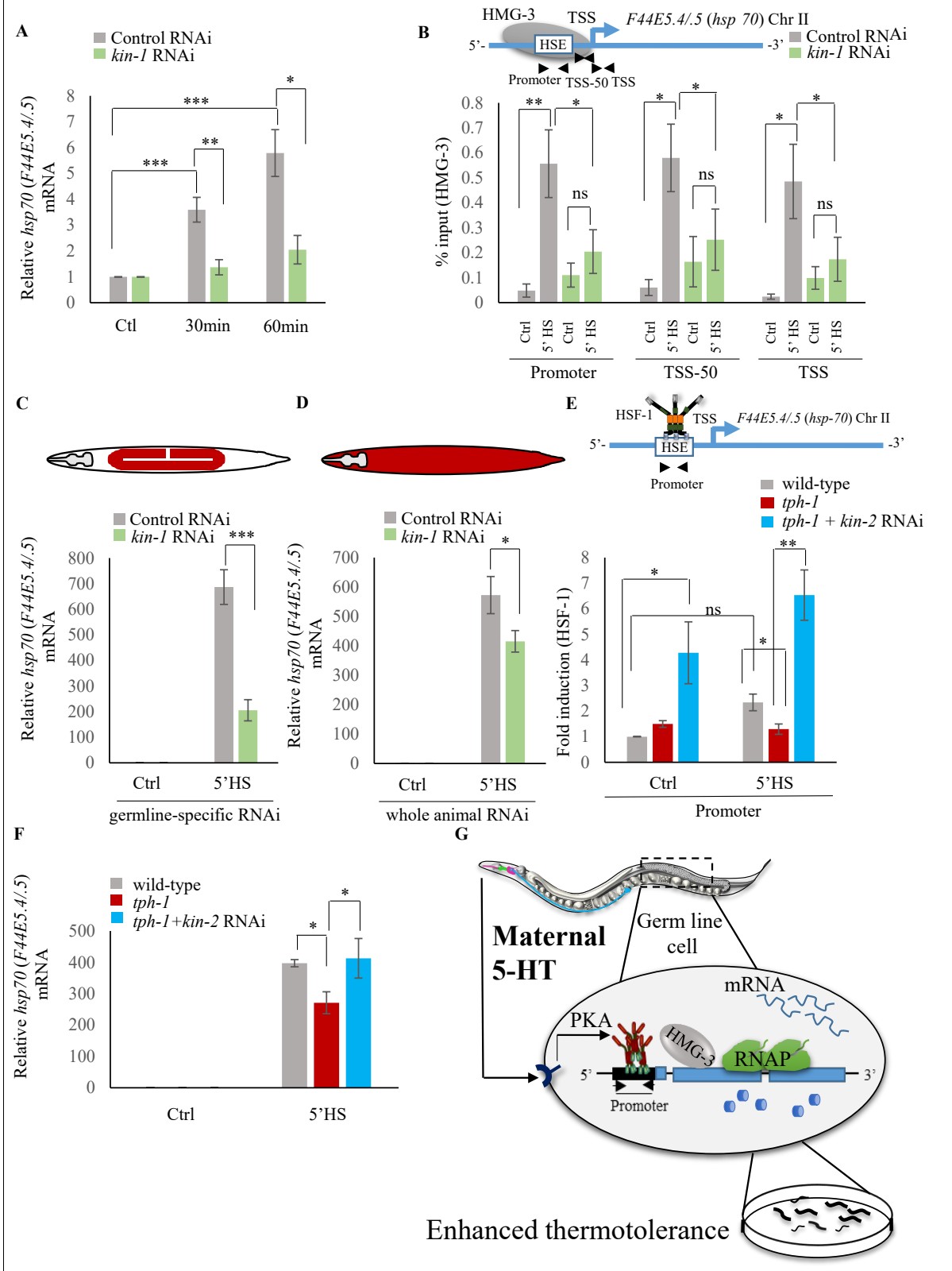

**Figure 8.** Serotonin-induced PKA-activation is a conserved signaling pathway that enables HSF-1 to recruit FACT in *C. elegans*. (**A**) *hsp70* (*F44E5.4/.5*) mRNA levels in control-RNAi treated and *kin-1*-RNAi treated animals at different time points following optogenetic stimulation of 5-HT release. mRNA levels are normalized to control RNAi and *kin-1* RNAi treated unstimulated animals respectively (n = 8 experiments). (**B**) Top: Schematic of *hsp70* (*F44E5.4/.5*) gene showing Promoter, TSS-50 and TSS region used to assess HMG-3 occupancy. Bottom: HMG-3 occupancy (% input) in wild-type

*Figure 8 continued on next page*

*Figure 8 continued*

animals subjected to control and *kin-1* RNAi following 5 min at 34°C, across Promoter, TSS-50 and TSS regions (n = 9 experiments). Specificity and efficiency of pull-down under control conditions was ascertained. (**C**) *hsp70* (*F44E5.4/.5*) mRNA levels in control and heat-shocked (*mkcSi13 [sun-1p::rde-1::sun-1 3'UTR + unc-119*(+)] *II; rde-1(mkc36) V*) animals that undergo germ-line specific RNAi following exposure to control RNAi or *kin-1* RNAi. HS: 5 min at 34°C (n = 4 experiments). (**D**) *hsp70* (*F44E5.4/.5*) mRNA levels in control and heat-shocked wild-type animals after they were subject to control and *kin-1* RNAi-mediated knockdown. HS: 5 min at 34°C (n = 4 experiments). (**E**) Top: Schematic of *hsp70* (*F44E5.4/.5*) promoter region assayed for HSF-1 occupancy. Bottom: HSF-1 occupancy in control and heat-shocked wild-type animals, *tph-1* mutant animals and *tph-1* mutant animals subjected to *kin-2* -RNAi. HS: 5 min at 34°C (n = 5 experiments). % input values were normalized to that in control wild-type animals not subjected to heat shock. (**F**) *hsp70* (*F44E5.4/.5*) mRNA levels in control and heat-shocked wild-type animals, *tph-1* mutant animals and *tph-1* mutant animals subjected to *kin-2* -RNAi. HS: 5 min at 34°C (n = 5 experiments). Data show Mean ± Standard Error of the Mean. *, $p<0.05$; **, $p<0.01$ ***, $p<0.001$; ns, non-significant. (**B**, **E**) ANOVA with Tukey's correction, A, C, D, F: paired Student's t-test). (**G**) Working model showing how maternal 5-HT protects future progeny. 5-HT released by neurons permeates the animal through the coelomic fluid to act on the germline, activate the PKA signal transduction pathway in a tissue-autonomous manner and enable HSF-1 to recruit FACT, displace nucleosomes in the germline, and accelerate the onset of protective gene expression. The online version of this article includes the following figure supplement(s) for figure 8:

**Figure supplement 1.** Serotonin-induced PKA-activation is a conserved pathway that enables HSF-1 to recruit FACT in *C. elegans*.

(*Taylor and Dillin, 2013*) and mitochondria (*Berendzen et al., 2016*) are controlled cell non-autonomously by the nervous system. However, the mechanism by which this occurs was not known; neither was it clear whether such regulatory control was conserved. Here, we show that 5-HT released from maternal neurons in *C. elegans* upon stress allows the information of the stress stimulus to be linked to the onset of protective HSF1-dependent transcription in germ cells, ensuring their survival upon fertilization and enhanced stress tolerance as larvae. 5-HT-mediates these effects by enabling HSF-1 to modify chromatin through the activity of the histone chaperone FACT and accelerate the onset of transcription by displacing histones. Thus, 5-HT release by neurons, in effect, sets the level of the physiological stimulus required to activate a transcriptional response amongst the germline nuclei. Remarkably, maternal 5-HT release upon stress causes an increase in *hsp70* mRNA levels in embryos and transgenerational stress tolerance in progeny. Given the role of neuronal 5-HT in modulating memory and learning these studies have wide-ranging implications for the effects on maternal experience on progeny physiology.

The exact mechanism by which embryos have more *hsp* mRNA was not explored; however, this is consistent with published observations that even if oocytes are not transcriptionally competent, pachytene nuclei can act as nurse cells to provide material to uncellularized oocytes (*Wolke et al., 2007*) to be subsequently utilized by embryos. It is possible that besides *hsps*, 5-HT also stimulates the packaging of other mRNAs into soon-to be fertilized embryos to promote their survival. This remains to be determined. In mammalian cells, 5-HT activates HSF1 through the 5-HT4 receptor. *C. elegans* do not possess a direct ortholog of 5-HT4 (*Chase and Koelle, 2007*; *Curran and Chalasani, 2012*) and the receptor required to activate PKA in the germline upon heat shock remains to be identified. One possible receptor is the SER-7, which although has not been localized to the germline, controls numerous aspects of egg-laying and oocyte maturation, acting through a $G\alpha_s$-coupled signaling pathways to promote PKA-dependent phosphorylation of target proteins (*Chase and Koelle, 2007*; *Cheng et al., 2008*; *Curran and Chalasani, 2012*).

Our data also show that for unknown reasons, fertilized *C. elegans* embryos are exquisitely vulnerable to even transient temperature fluctuations. Lowering the threshold of transcription onset in the germline by 5-HT is therefore critical to protect development, and indeed the germline is amongst the first tissues to express protective *hsp70s*. Across tissues and during development, transcriptional responses display characteristic dynamics and thresholds of activation that are linked to their biological function (*Batulan et al., 2003*; *Kotas and Medzhitov, 2015*; *Vihervaara et al., 2018*; *Zhou et al., 2018*). In addition transcriptional responses change, dynamically and swiftly, with the continued presence of the stressor (*de Nadal et al., 2011*). Here, we provide a molecular mechanism by which thresholds for activation of transcription upon stress can be set to different levels in different tissues *in vivo*.

The essential aspects of the 5-HT signaling pathway are conserved in mammalian neurons. The involvement of PKA in 5-HT dependent HSF1 activation is intriguing given the role of both 5-HT and PKA in cellular plasticity. It is therefore tempting to speculate that the ability to modulate transcriptional dynamics through modifying chromatin accessibility may be a more general function of 5-HT

in neurodevelopment and as a neuromodulator, allowing it to steer developmental timing and neuronal activity (*Bonnin et al., 2011*; *Dowell et al., 2019*; *Iwanir et al., 2016*; *McKay, 2011*; *Puscheck et al., 2015*). In addition, this ability to functionally activate HSF1 in mammalian neurons and human cells, in the absence of proteotoxicity, through activation of 5HT4 receptors could have implications for the treatment of neurodegenerative diseases where HSF1 is protective (*Lee et al., 1995*; *Li et al., 2017*; *Westerheide and Morimoto, 2005*).

Why might 5-HT, an abundant neuromodulator and signaling molecule that portends growth, also modulate stress responsiveness of germ cells? The answer to this question may be limited by our understanding of what precisely constitutes 'stress' for different cells. Germ cells typically consist of 'poised' chromatin (*Choate and Danko, 2016*) bearing both activation and repressive histone marks, which potentially can resolve into growth-related and 'active', or stress-related and 'repressed' antagonistic gene expression programs (*López-Maury et al., 2008*). Across the animal kingdom, 5-HT release can signal stress or growth (*Berger et al., 2009*; *Chaouloff, 2002*). We postulate that the ability of 5-HT to modulate chromatin accessibility in response to environmental input may allow it to function as a switch at the nexus of these essential programs.

# Materials and methods

## Key resources table

| Reagent type (species) or resource | Designation | Source or reference | Identifiers | Additional information |
|---|---|---|---|---|
| Strain, strain background (*Mus musculus*) | Wild type C57BL/6 | Envigo | | Cortical neuron preparations were made from P0 pups from timed-pregnant C57BL/6 mice |
| Genetic reagent (*C. elegans*) | N2 Wild type | Caenorhabditis Genetic Center | N2, *C. elegans* var *Bristol* | |
| Genetic reagent (*C. elegans*) | *hsf-1::3X flag I* | Prahlad lab, University of Iowa | CRISPR insertion of 3X FLAG at C' of *hsf-1* | This paper (Materials and methods) |
| Genetic reagent (*C. elegans*) | *tph-1 (mg280) II* | Caenorhabditis Genetic Center | Strain Name: MT15434 | |
| Genetic reagent (*C. elegans*) | *hsf-1(sy441)I* | Caenorhabditis Genetic Center | Strain Name: PS3551 | |
| Genetic reagent (*C. elegans*) | *lite-1(ce314); ljIs102 [tph-1;;ChR2::YFP; unc-122::gfp]* | Caenorhabditis Genetic Center | Strain Name: AQ2050 | |
| Genetic reagent (*C. elegans*) | *hmg-3(bar24 [hmg-3::3xHA]) I* | Gift from Dr. Baris Tursun, Max Delbrück Center (MDC) | Strain Name: BAT1560 | |
| Genetic reagent (*C. elegans*) | *glp-4(bn2) I* | Caenorhabditis Genetic Center | Strain Name: SS104 | |
| Genetic reagent (*C. elegans*) | *rrf-1(pk1417) I* | Caenorhabditis Genetic Center | Strain Name: NL2098 | |
| Genetic reagent (*C. elegans*) | *mkcSi13 II; rde-1 (mkc36) V* | Caenorhabditis Genetic Center | Strain Name: DCL569 | |
| Genetic reagent (*C. elegans*) | *rde-1(ne219) V; kbIs7* | Caenorhabditis Genetic Center | Strain Name: VP303 | |
| Genetic reagent (*C. elegans*) | *tph-1 (mg280) II; hsf-1::flag I* | Prahlad lab, University of Iowa | Cross made with *hsf-1 ::3Xflag (I)* strain and *tph-1 (II)* | This paper (Materials and methods) |
| Genetic reagent (*C. elegans*) | *tph-1 (mg280) II; hmg-3 (bar24[hmg-3::3xHA]) I* | Prahlad lab, University of Iowa | Cross made with FLAG tagged train and *tph-1* | This paper (Materials and methods) |
| Cell line (*Homo-sapiens*) | NTERA-2 cl.D1 (NT2 cell line) | Originally from Stratagene | | A gift from Dr. Christopher Stipp, University of Iowa (PMID:10806098) |

*Continued on next page*

*Continued*

| Reagent type (species) or resource | Designation | Source or reference | Identifiers | Additional information |
|---|---|---|---|---|
| Antibody | Mouse monoclonal anti-FLAG | Sigma | Catalog No. F1804 | IF (1:100), WB (1:1000) |
| Antibody | Mouse monoclonal anti-RNA polymerase II | BioLegend | Catalog No. 664906 | ChIP (5 µl per sample) |
| Antibody | Rabbit polyclonal anti-Histone H3 | Abcam | Catalog No. ab1791 | ChIP (2 µl per sample) |
| Antibody | Rabbit polyclonal anti-HA | Abcam | Catalog No. ab9110 | ChIP (5 µl per sample), WB (1:1000) |
| Antibody | Rabbit polyclonal anti-HSF1 | Cell Signaling Technology | Catalog No. 4356 | IF (1:100), WB (1:1000) |
| Antibody | Rabbit monoclonal anti-HSF1(S320) | Abcam | Catalog No. ab76183 | WB (1:1000) |
| Antibody | Mouse monoclonal anti-Hsp70 | Gift from Dr. Richard Morimoto, Northwestern University | Clone: 3A3 | WB (1:1000) |
| Antibody | Mouse monoclonal anti-α-tubulin | DSHB, University of Iowa | Catalog No. AA4.3 | WB (1:1000) |
| Antibody | Donkey anti-mouse Cy3 | Jackson ImmunoResearch Laboratories | Catalog No. 715-165-150 | IF (1:1000) |
| Antibody | AlexaFluor 488-conjugated goat anti-rabbit IgG | Invitrogen | Catalog No. A-11008 | IF (1:500) |
| Antibody | Sheep anti-mouse IgG | Rockland Immunochemicals | Catalog No. 610-631-002 | WB (1:10,000) |
| Antibody | Goat anti-rabbit IgG | Invitrogen | Catalog No. A21109 | WB (1:10,000) |
| Sequence-based reagent | siRNA targeting human *HSF1* | Santa Cruz Biotechnology | Catalog No. sc-35611 | These are pools of 3–5 target-specific 19–25 nucleotide sequences designed by the company |
| Sequenced-based reagent | siRNA targeting human *SUPT16H* | Santa Cruz Biotechnology | Catalog No sc-37875 | These are pools of 3–5 target-specific 19–25 nucleotide sequences designed by the company |
| Sequenced-based reagent | Control siRNA | Santa Cruz Biotechnology | Catalog No sc-37007 | These are pools of 3–5 target-specific 19–25 nucleotide sequences designed by the company |
| Commercial assay or kit | Mouse anti-FLAG M2 magnetic bead | Sigma | Catalog No. M-8823 | ChIP (15 µl per sample) |
| Commercial assay or kit | Protein A/G magnetic bead | Pierce | Catalog No. 88802, Pierce | ChIP (15 µl per sample) |
| Commercial assay or kit | iScript cDNA Synthesis Kit | Bio-Rad | Catalog No. 170–8891 | |
| Commercial assay or kit | SYBR Green Master Mix | Roche | Catalog No. 04887352001 | |
| Commercial assay or kit | Lipofectamine LTX Plus reagent | Thermo Fisher Scientific | Catalog No. 15338030 | |
| Commercial assay or kit | ChIP DNA purification kit | Zymo Research | Catalog No. D5205 | |
| Chemical compound, drug | 5-HT | Sigma | Catalog No. 85036 | |

*Continued on next page*

*Continued*

| Reagent type (species) or resource | Designation | Source or reference | Identifiers | Additional information |
|---|---|---|---|---|
| Chemical compound, drug | Sumatriptan succinate (5-HT1 receptor agonist) | Sigma | Catalog No. S1198 | |
| Chemical compound, drug | DOI hydrochloride (5-HT2A receptor agonist) | Sigma | Catalog No. D101 | |
| Chemical compound, drug | BIMU8 hydrate (5-HT4 receptor agonist) | Sigma | Catalog No. B4063 | |
| Chemical compound, drug | ST1936 (5-HT6 receptor agonist) | Sigma | Catalog No. SML0260 | |
| Chemical compound, drug | PKA inhibitor H89 | Sigma | Catalog No. B1427 | |
| Software, algorithm | ImageJ (FIJI) | NIH | | Quantification of fluorescence intensity |
| Software, algorithm | LAS X | Leica Microsystems | | Confocal imaging and analysis |

## *C. elegans* strains

Most *C. elegans* strains were obtained from Caenorhabditis Genetics Center (CGC, Twin Cities, MN). The BAT1560 strain was a kind gift from Dr. Baris Tursun, Max Delbrück Center (MDC) (*Kolundzic et al., 2018*). The HSF-1::FLAG strain was created using CRISPR/Cas9.

| Strain name | Genotype | Source |
|---|---|---|
| HSF-1::FLAG | *hsf-1::3X flag I* | Prahlad lab |
| Wild-type | Bristol N2 | CGC |
| MT15434 | *tph-1 (mg280) II* | CGC |
| PS3551 | *hsf-1(sy441)I* | CGC |
| AQ2050 | lite-1(ce314); *ljIs102* *[tph-1;;ChR2::YFP; unc-122::gfp]* | CGC |
| BAT1560 | *hmg-3(bar24 [hmg-3::3xHA]) I* | Dr. Baris Tursun, Max Delbrück Center (MDC) |
| SS104 | *glp-4(bn2) I* | CGC |
| NL2098 | *rrf-1(pk1417) I* | CGC |
| DCL569 | *mkcSi13 II; rde-1(mkc36) V* | CGC |
| VP303 | *rde-1(ne219) V; kbIs7* | CGC |

## Generation of *hsf-1*::FLAG

CRISPR/Cas9 was used to create *C. elegans* strains where the endogenous *hsf-1*(I) gene was tagged at the C-terminus with a 3X FLAG sequence to create *HSF-1::FLAG animals (hsf-1::3Xflag (I))*. Individual adult worms were injected on 3% agarose pads with the injection mix detailed below. Following injection, animals were singled onto NGM plate. Plates were screened for the *rol* or *dpy* phenotypes created by the co-CRISPR marker *dpy-10* (*Paix et al., 2016*). One hundred animals with a DPY or Roller phenotype were isolated as F1s and screened for the FLAG insertion by PCR. Three days later, single wild-type F2 offspring from plates with Dpy and/or Rol offspring were singled, screened for homozygosity of the FLAG insertion by PCR, and sequenced. The Cas9 enzyme, ultramer oligonucleotides, tracrRNA, and crRNAs were obtained from IDT.

| Tracr RNA | 30 μM |
|---|---|
| crRNA- dpy10 | 5 μM |
| crRNA- TARGET | 25 μM |
| IDT Cas9 Enzyme | 12.2 μM |
| dpy-10 ssODN | 0.5 μM |
| Target ssODN | 5 μM |

The sequences of the crRNA and the ssODN for hsf-1 are: crRNA-C': AAGTCCATCGGATCCTAA TT ssODNhsf-1-C: TCCCACATTCACCGGCTCTTCGTACTCCAAGTCCATCGGATCCTAATTTGG TTgactacaaagaccatgacggtgattataaagatca tgaTatcgaTtacaaggatgacgatgacaagTAAttattgatttttttttttgaacgtttagctcaaaattcctctc.

## Generation of *tph-1*; *hsf-1*::FLAG and *tph-1*; *hmg-3*::HA

The *tph-1 (mg280) II* strain was crossed into the *hsf-1::3Xflag (I)* strain or *hmg-3::HA (I)* and verified by PCR.

## Growth conditions of *C. elegans* strains

All strains except *glp-4(bn2) I* were grown and maintained at 20°C; *glp-4 (bn2) I* worms were grown and maintained at 15°C (permissive temperature). For the experiments involving *glp-4 (bn2) I*, glp-4 *(bn2) I* eggs or wild-type eggs were raised at the permissive temperature (15°C), or shifted to 25°C after they were laid at 15°C until animals were day-1 adults. Animals were grown and maintained at low densities in incubators under standard conditions by passaging 8–10 L4s onto nematode growth media (NGM) plates and, 4 days later, picking L4 animals onto fresh plates for experiments. Animals were fed *Escherichia coli* OP50 obtained from CGC that were seeded onto culture plates 2 days before use. The NGM plates were standardized by pouring 8.9 ml of liquid NGM per 60 mm plate weighed before use. Plates had an average weight of $13.5 \pm 0.2$ g. Any plates that varied from these measurements were discarded. Ambient temperature was maintained at 20°C to 22°C and carefully monitored throughout the experimental procedures. All animals included in the experiments, unless otherwise stated, were 1-day-old hermaphrodites that were age-matched either by (a) bleaching and starting the experiment after 75–78 hr or (b) picking as L4 juveniles 24 to 26 hr before the start of the experiment.

## Mammalian cell culture

Dr. Christopher Stipp, University of Iowa, gifted NTERA-2 cl.D1 (also known as NT2) cells. For regular maintenance, cells were cultured in DMEM (Life Technologies) supplemented with 10% fetal bovine serum (Life Technologies), 2 mM L-glutamine and 100 U/ml penicillin and 100 μg/ml streptomycin. Cells were maintained at 37°C in 5% $CO_2$ atmosphere under humidified conditions. Cell were passaged by splitting them (1:4) when cell confluence reached ~90%. All cells used were between passage numbers 15 and 20. Cells were routinely checked for mycoplasma contamination.

## Mouse strain and cortical neuron culture

Cortical neuron cultures were performed essentially as described previously (*Garrett et al., 2012*; *Keeler et al., 2015*) using P0 pups from timed-pregnant C57BL/6 mice (RRID:MGI_5658456, Envigo). Briefly, cortices were dissected, meninges were removed, and ~1 mm² pieces were digested in an enzyme solution (papain, 10 units/ml) $2 \times 20$ min. The tissue was rinsed with increasing concentrations of trypsin inhibitor followed by plating medium (Basal Medium Eagle, 5% fetal bovine serum, Glutamax (Invitrogen), N2 supplements (Invitrogen), and penicillin/streptomycin). Cells were plated onto 12 mm round German cover glass coated with Matrigel (Corning) at a density of ~250,000 cells per coverslip. After 4 hr and every 2–3 days subsequently, 50% of the medium was changed to fresh Neurobasal supplemented with Glutamax, GS21 supplements (AMSBIO), and penicillin/streptomycin.

## Heat shock of worms

NGM plates (8.9 ml liquid NGM/plate, weight 13.5 ± 0.2 g) were seeded with 300 µl OP50 in the center and allowed to dry for 48 hr. Either L4 hermaphrodites were passaged on to these plates or worms were bleach- hatched on to these plates and allowed to grow to Day1 adults. All heat shock experiments were performed with 1-day-old gravid animals. To induce heat shock response in *C. elegans*, NGM plates containing 1-day-old animals were parafilmed and immersed in water bath (product no. ITEMP 4100 H21P 115V, Fischer Scientific, Pittsburgh, PA) pre-warmed to 34°C, for indicated times (5 or 15 min). When required, animals were recovered in 20°C incubators following heat shock, under standard condition after the parafilm was removed. Animals were harvested immediately following heat shock, or following recovery, by rapidly washing them off the plates in sterile water or the appropriate buffer (for ChIP-qPCR and RNA seq experiments) or by picking into 1.5 ml tubes (optogenetics).

## Transfection of mammalian cells

NT2 cells were transfected with Lipofectamine LTX Plus reagent (catalog no. 15338030, Thermo Fisher Scientific) according to manufacturer's protocol.

## Exogenous 5-HT treatment of worms

As described previously (*Ooi and Prahlad, 2017*), a 5-HT (catalog no. 85036, Sigma-Aldrich) stock solution of 10 mM was made in sterile water, filter-sterilized and then diluted to 2 mM before use. This solution (or sterile water as control) was dropped onto the surface of OP50 bacterial lawns (such that the lawns were fully covered in 5-HT) on NGM plates and allowed to dry for ~2 hr at room temperature. Day one adult animals were placed onto the 5-HT-soaked OP50 bacterial lawns.

## 5-HT treatment of cells or mouse cortical neurons

Cells were seeded at $1 \times 10^5$ cells/ml density the day before the experiment. Cell density influenced the experimental outcome and therefore cell numbers were maintained by counting in the hemocytometer. Since regular serum contains 5-HT, cells were grown in presence of dialyzed fetal bovine serum (Thermo Fisher Scientific) for at least 24 hr prior to all experiments. A 5-HT (catalog no. 85036, Sigma-Aldrich) stock solution of 10 mM was made in sterile water. Cells were incubated at 37°C with different concentrations of 5-HT (or sterile water as control) for different time periods as mentioned in the figures/figure legends and harvested for subsequent assays. For mouse cortical neurons, on the 11th day in vitro, each coverslip containing ~250,000 cells was incubated with different concentrations of 5-HT (or sterile water as control) for different time points. Following this incubation, medium was removed and cultures were immediately harvested for RNA preparation.

## Treatment of mammalian cells with 5-HT agonists

Stock solutions of Sumatriptan succinate (5-HT1 receptor agonist; catalog no. S1198, Sigma-Aldrich), DOI hydrochloride (5-HT2A receptor agonist; catalog no. D101, Sigma-Aldrich), BIMU8 hydrate (5-HT4 receptor agonist; catalog no. B4063, Sigma-Aldrich) and ST1936 (5-HT6 receptor agonist; catalog no. SML0260, Sigma-Aldrich) were made in sterile water or DMSO (also used as control). Cells were grown overnight at described densities in presence of dialyzed fetal bovine serum, incubated with different concentrations of 5-HT agonists for indicated time periods (specified in figures and figure legends) and immediately harvested for experiments. Mouse cortical neurons were treated with BIMU8 hydrate following the protocol for exogenous 5-HT described above.

## Treatment of cells with PKA inhibitor (H89)

Cells were grown overnight in presence of dialyzed fetal bovine serum and then treated with 10 µM H89 (catalog no. B1427, Sigma-Aldrich) for 2 hr. and followed by treatment with 5-HT4 agonist (or control).

## Bleach hatching

*C. elegans* populations contained a large number of gravid adults were selected by picking maintenance plates 5 days after the passage of L4s, as described above. Animals were washed off the plates with 1X PBS and the worms were pelleted by centrifuging at 5000 rpm for 30 s. The PBS was

removed carefully, and worms were gently vortexed in presence of bleaching solution (250 μl 1N NaOH, 200 μl standard bleach, 550 μl sterile water) until all the worm bodies were dissolved (approximately 5–6 min). The eggs were pelleted by centrifugation (5000 rpm for 45 s) and bleaching solution was removed. Eggs were washed with sterile water three times and then counted. Care was taken to ensure that all the embryos hatched following this treatment. The eggs were seeded on fresh OP50 or RNAi plates (~100 eggs/plate for gene expression analysis and ~200 eggs/plate for chromatin immunoprecipitation) and allowed to grow as day-1-adults under standard condition (20° C).

## RNA interference

RNAi experiments were conducted using the standard feeding RNAi method. Bacterial clones expressing the control (empty vector pL4440) construct and the dsRNA targeting different *C. elegans* genes were obtained from the Ahringer RNAi library (*Kamath and Ahringer, 2003*) now available through Source Bioscience (https://www.sourcebioscience.com/errors?aspxerrorpath=/products/life-science-research/clones/rnai-resources/c-elegans-rnai-collection-ahringer/). *kin-1* RNAi construct was obtained from Dharmacon (catalog no. RCE1182-202302363). All RNAi clones used in experiments were sequenced for verification before use. For RNAi experiments, RNAi bacteria with empty (pL4440 vector as control) or RNAi constructs were grown overnight in LB liquid culture containing ampicillin (100 μg/ml) and tetracycline (12.5 μg/ml) and then induced with IPTG (1 mM) for 2 hr before seeding the bacteria on NGM plates supplemented with ampicillin (100 μg/ml), tetracycline (12.5 μg/ml) and IPTG (1 mM). Bacterial lawns were allowed to grow for 48 hr before the start of the experiment. RNAi-induced knockdown was conducted by (a) dispersing the bleached eggs onto RNAi plates or (b) feeding L4 animals for 24 hr (as they matured from L4s to 1-day-old adults) or (c) feeding animals for over one generation, where second-generation animals were born and raised on RNAi bacterial lawns (*hsf-1*). RNAi-mediated knockdown was confirmed by scoring for known knock-phenotypes of the animals subject to RNAi (slow and arrested larval growth as well as larval arrest at 27°C for *hsf-1* RNAi; dumpy adults for *kin-2* RNAi). *rrf-1(pk1417) I* (NL2098) and *mkcSi13 II; rde-1(mkc36) V* (DCL569) worms were used for germline-specific RNAi experiments whereas *rde-1(ne219) V; kbIs7* (VP303) worms were used for intestine-specific RNAi experiments. These worms were grown in control and *hsf-1* RNAi plates for two generations as mentioned above and day-1 adults were used for heat shock experiments. For germline-specific knockdown of *kin-1* and *kin-2*, *mkcSi13 II; rde-1(mkc36)* worms were bleach hatched on control RNAi or *kin-1/kin-2* RNAi plates and experiments were performed with 1-day-old animals.

## Knockdown of mammalian HSF1 and SUPT16H by siRNA

Control siRNA and siRNA targeting human *HSF1* and *SUPT16H* (SPT16) were procured from Santa Cruz Biotechnology Inc, USA (catalog no. sc-37007, sc-35611 and sc-37875 respectively) and NT2 cells were transfected with Lipofectamine LTX Plus reagent according to manufacturer's protocol. All experiments were performed 48 hr. after transfection and knockdown of endogenous HSF1 and *sp16* was confirmed by western blotting or qRT-PCR, respectively. The protein levels were quantified using ImageStudio (LI-COR).

## Optogenetic activation of serotonergic neurons

Optogenetic experiments were performed according to previously published methods as per the requirements of the experiment (*Ooi and Prahlad, 2017*; *Tatum et al., 2015*). Briefly, experimental plates (ATR+) were made from 100 mM ATR (product no. R2500, Sigma-Aldrich) stock dissolved in 100% ethanol and then diluted to a final concentration of 2.5 mM into OP50 or L4440 or *kin-1* or *hmg-3* RNAi bacterial culture and 200 μl was seeded onto a fresh NGM plate. Control (ATR-) plates were seeded at the same time with the same culture without adding ATR. All plates were allowed to dry overnight in the dark before use. The *C. elegans* strain AQ2050 was used for this experiment. L4s were harvested on to ATR+ and ATR- plates and the experiment was carried out with day one adults. All plates were kept in the dark and animals were allowed to acclimatize to room temperature (20°C to 22°C) for about 30 min. before starting the experiment. Animals were illuminated with blue light for 30 s at a 6.3X magnification using an MZ10 F microscope (Leica) connected to an EL6000 light source (Leica) and harvested at different time points as indicated in Trizol and snap-

frozen immediately in liquid nitrogen for RNA extraction. Optogenetic 5-HT release during light stimulation was confirmed by measuring pharyngeal pumping rates.

Single-molecule fluorescence in situ hybridization (smFISH) smFISH probes were designed against the worm *hsp70* (*F44E5.4/.5*) gene by using the Stellaris FISH Probe Designer (Biosearch Technologies Inc) available online at http://www.biosearchtech/com/stellarisdesigner. The fixed worms were hybridized with the *F44E5.4/.5* Stellaris FISH Probe set labeled with Cy5 dye (Biosearch Technologies Inc) following the manufacturer's protocol. About 20 wild-type (N2) day one worms per condition (control and 34°C heat shock for 5 and 15 min) were harvested by picking off plates immediately after heat exposure into 1X RNase-free phosphate-buffered saline (PBS) (catalog no. AM9624, Ambion), fixed in 4% paraformaldehyde, and subsequently washed in 70% ethanol at 4°C for about 24 hr to permeabilize the animals. Samples were washed using Stellaris Wash Buffer A (catalog no. SMF-WA1-60, Biosearch Technologies Inc), and then the hybridization solution (catalog no. SMF-HB1-10, Biosearch Technologies Inc) containing the probes was added. The samples were hybridized at 37°C for 16 hr, after which they were washed three times with Wash Buffer A and then incubated for 30 min in Wash Buffer A with DAPI. After DAPI staining, worms were washed with Wash Buffer B (catalog no. SMF-WB1-20, Biosearch Technologies Inc) and mounted on slides in about 16 µl of Vectashield mounting medium (catalog no. H-1000, Vector Laboratories). Imaging of slides was performed using a Leica TCS SPE Confocal Microscope (Leica) using a 63X oil objective. LAS AF software (RRID:SCR_013673, Leica) was used to obtain and view z- stacks.

## Single-molecule fluorescence in situ hybridization (smFISH) quantification

Wild-type control animals kept at 20°C, and animals harvested immediately following heat shock at 34°C for 5 and 15 min were prepared for smFISH. Following hybridization of the probes, confocal stacks (z = 0.35 µm) of whole animals were obtained using the 'tile scan' feature of the Leica TCS SPE Confocal Microscope. Z-projections of the head, the first two cells of the intestine, and the pachytene region of the germline were used for quantification. The number of spots that result from the hybridization of the *hsp70* (*F44E5.4/.5*) probe were counted using ImageJ (RRID:SCR_003070). Stacks were collapsed to generate a composite image, split to the respective single channel images using the Split Channels command under 'Image/Color/Split Channels'. The image was thresholded using 'Image/Adjust/Threshold' Plugin using the MaxEntropy feature which is an automatic thresholding method that uses the entropy of the histogram of the image to generate the threshold. The number of puncta were counted manually using the multi-point tool. This method worked optimally for counting the number of smFISH signals which remained discrete. However, because puncta in nuclei in the head and intestine began to merge with each other in projected confocal slices by 15 min of heat shock, we counted them in separate planes. Separately the background fluorescence in control animals and in the intestine were measured in a segmented region of the image using the Analyze/Measure in ImageJ, to ensure that we were not missing diffuse increases in signal between the different conditions. These methods do not resolve individual RNA molecules, but can be used to quantify the number of discrete point-signals from the hybridization and compare relative differences. Subsequently a program in Windows C++ was written and is currently used to streamline the counting. This has confirmed our manually curated data, and the source code of this program is available upon request.

## Immunofluorescence staining of dissected gonads

Immunostaining of dissected gonads of *C. elegans* was performed to visualize HSF-1::FLAG in the germline. The procedure was conducted as described earlier (*Ooi and Prahlad, 2017*). Day-1 wild-type adults harboring HSF-1::FLAG and *tph-1* HSF-1::FLAG worms were picked into 15 µl of 1X PBS (pH7.4) on a coverslip, and quickly dissected with a blade (product no. 4–311, Integra Miltex). A charged slide (Superfrost Plus, catalog no. 12-550-15, Thermo Fisher Scientific) was then placed over the coverslip and immediately placed on a pre-chilled freezing block on dry ice for at least 5 min. The coverslip was quickly removed, and the slides were fixed in 100% methanol (−20°C) for 1 min. and then fixed in 4% paraformaldehyde, 1X PBS (pH7.4), 80 mM HEPES (pH 7.4), 1.6 mM MgSO$_4$ and 0.8 mM EDTA for 30 min. After rinsing in 1X PBST (PBS with Tween 20), slides were blocked for 1 hr in 1X PBST with 1% BSA and then incubated overnight in 1:100 mouse anti-FLAG (catalog no.

F1804, RRID:AB_262044, Sigma Aldrich) antibody. The next day, slides were washed and then incubated for 2 hr. in 1:1000 donkey anti-mouse Cy3 (code no. 715-165-150, RRID:AB_2340813, Jackson ImmunoResearch Laboratories) before they were washed and incubated in DAPI in 1X PBST and then mounted in 10 µl of Vectashield mounting medium (catalog no. H-1000, Vector Laboratories) and imaged as described above.

## Immunofluorescence staining of NT2 cells

NT2 cells grown overnight on coverslips ($1 \times 10^5$ cells/ml density) in presence of dialyzed fetal bovine serum were fixed in 4% paraformaldehyde in PBS at RT for 10 min. Fixed cells were permeabilized with 0.1% Triton-X-100 in PBS at 37°C for 5 min, blocked with 1% BSA in PBS at 37°C for 30 min, and incubated with rabbit anti-HSF1 antibody (1:100 dilution) (catalog no. 4356, RRID:AB_2120258, Cell Signaling Technology) for 2 hr. After washing, cells were incubated with AlexaFluor 488-conjugated goat anti-rabbit IgG (H+L) (catalog no. A-11008, RRID:AB_143165, Invitrogen) for 2 hr. After washing, coverslips were mounted in Vectashield mounting medium containing DAPI and imaged as mentioned earlier. Images were collected using a Leica Confocal SPE8 microscope using a 63 × numerical aperture 1.42 oil-immersion objective lens. The relative intensity of HSF1 in the nuclei of control NT2 cells, and cells treated with BIMU8 in the presence or absence of H89 was quantified from the projections of confocal z-stacks using ImageJ. Background signal was subtracted from each of the projections and the mean intensity for regions corresponding to nuclei of the cells was determined. The average of 25 cells was used to determine the mean intensity for HSF1 staining.

## Assays to evaluate progeny survival following heat shock

### Progeny survival following 5 min and 15 min maternal heat-shock

N2 and *tph-1* L4s were picked on fresh OP50 plates the day before the experiment. After 24–26 hr, 1-day-old animals were either heat shocked at 34°C for 5 or 15 min in the water bath or left untreated (control). Heat-shocked animals were either (a) moved to fresh OP50 plates to lay eggs for a 2 hr duration immediately after heat shock (0–2 hr. embryos) or (b) allowed to recover in an incubator at 20°C for 2 hr, and then moved to fresh OP50 plates and allowed to lay eggs for a 2 hr duration (2–4 hr.). Control embryos were those laid by non-heat shocked animals from the same 2 hr duration. For all experiments except those processed for mRNA, embryos scored were from five worms per plate (2–3 plates per experiment). To score viable embryos, the number of eggs laid were counted, embryos were allowed to hatch at 20°C incubator, and the number of live progeny were scored 48 hr later. We ascertained that these larvae subsequently grew into adults.

### Progeny survival following heat-shock following RNAi-induced knockdown in parents, or 5-HT treatment of parents

When RNAi treatment was required, the parents were bleach hatched on fresh RNAi plates and allowed to grow under standard condition (20°C). Animals on Day-1 of adulthood were then transferred onto fresh RNAi plates, subjected to heat-shock or used as controls as described in order to calculate the percentage of live progeny. For assaying rescue of *tph-1* embryonic viability by 5-HT treatment, *tph-1* mutant animals were transferred to 5-HT plates made as described above, immediately heat-shocked for 5 min at 34°C, allowed to recover for 2 hr on the same plate, and then transferred to fresh OP50 plates without 5-HT to lay eggs for 2 hr. This was because the rate of transit of exogenous 5-HT through the animal is poorly understood.

### Survival of homozygous and heterozygous progeny

To assess maternal contribution, five *tph-1* hermaphrodites (L4s) were allowed to mate with 10 wild-type (N2) males for 26 hr. Wild-type hermaphrodites (L4) were also allowed to mate with wild-type males for 26 hr in similar numbers to control for any effects of mating. Mating was ascertained by counting, post-hoc, the numbers of male progeny laid by these hermaphrodites and ensuring they were ~50% male. The mated hermaphrodites were heat shocked at 34°C for 5 min, the males removed, and hermaphrodites allowed to recover for 2 hr at 20°C, and then transferred to new OP50 plates to lay eggs for a 2 hr interval. The embryos laid by these hermaphrodites were scored for viability as described above. The hermaphrodites were then transferred to new plates and their

male progeny were counted so as to ascertain they had indeed mated. Unmated wild-type and *tph-1* hermaphrodites were also heat shocked at the same time. Mated and unmated wild-type and *tph-1* animals that were not subjected to heat shock were used as control.

## Survival assay to determine the contribution of heat-shocked sperm

Wild-type day-1 males were heat shocked at 34°C for 5 min and then transferred onto plates containing L4 wild-type hermaphrodites and allowed to mate for 26 hr. Mating was ascertained by counting, post-hoc, the numbers of male progeny laid by these hermaphrodites and ensuring they were ~50% male. The gravid 1-day-old hermaphrodites were then heat shocked at 34°C for 5 min, the males removed, and hermaphrodites transferred immediately onto new OP50 seeded plates to lay eggs for a duration of 2 hr. The hermaphrodites were then transferred to new plates and their male progeny counted so as to ascertain they had indeed mated. Percent viability of embryos was calculated as mentioned earlier.

## Progeny survival following a prolonged heat exposure

Control and heat-shocked (34°C for 5 min), wild-type and *tph-1* day-1 animals were allowed to lay eggs for 2–4 hr post-heat shock and then all animals were taken off from the plates. After 48 hr, the numbers of progeny that hatched was calculated as described above, and the progeny were then subjected to a prolonged (3 hr.) heat exposure of 34°C. This condition was chosen after prior experiments to titrate death of control, non-heat shocked progeny to ~50% to prevent ceiling effects. The percent larvae that survived the prolonged heat shock was scored 24 hr later.

## RNA-sequencing and data analysis

a. RNA isolation, library preparation and sequencingAge synchronized day one adult wild-type, *tph-1(mg280)*II and *hsf-1(sy441)*I animal, upon heat-shock or control conditions, were harvested for RNA extraction. Total RNA was extracted from biological triplicates. Sample lysis was performed using a Tissuelyser and a Trizol-chloroform based method was used in conjunction with the Zymo RNA Clean and Concentrator kit to obtain RNA. The Illumina TruSeq stranded mRNA kit was used to obtain stranded mRNA via Oligo-dT bead capture, and cDNA libraries were prepared from 500 ng RNA per sample. Use of stranded cDNA libraries have been shown to maximize the accuracy of transcript expression estimation, and subsequent differential gene expression analysis (*Zhao et al., 2015*). Each sample was multiplexed on 6 lanes of the Illumina HiSeq 4000 sequencer, generating $2 \times 150$ bp paired end reads, with about 43 to 73 million reads per sample.

b. RNA-seq analysis
FASTQC (RRID:SCR_014583) was used to evaluate the quality of the sequences. Sequence reads were trimmed of adapters contamination and 20 base pairs from the 5' and 3' ends by using Trim Galore Version 0.6.0 (www.bioinformatics.babraham.ac.uk/projects/trim_galore/) (RRID:SCR_011847). Only reads with a quality higher than Q25 were maintained. HISAT2 (*Pertea et al., 2016*) (RRID:SCR_015530) was used to maps the trimmed reads to the *C. elegans* genome release 35 (WBcel235). On average, 99.4% of the reads mapped to the reference genome. Assemblies of the sequences were done with StringTie (*Pertea et al., 2015*) (RRID:SCR_016323) using the gene annotation from Ensembl WBcel235 (*Zerbino et al., 2018*). DESeq2 (*Love et al., 2014*) (RRID:SCR_015687) was used to identify the genes differentially expressed between the samples. Genes with low read counts (n < 10) were removed from the DESeq2 analysis. Genes with a False Discovery Rate < 0.01 were considered significant. The genes selected for the heatmaps were the genes with significant differences in the wild type control vs wild type heat shock samples. If these genes were not significant in *sy441* control vs *sy441* heat shock or *tph-1* control vs *tph-1* heat shock comparisons, the $\log_2$ foldchange values were adjusted to 0. Principal component analysis (PCA) and pairwise distance analysis (sample-to-sample) were performed by using normalized counts coupled with the variance stabilization transformation (VST). The PCA was done using the top 100 genes with the highest variance in read counts, the pairwise distance analysis was done using the complete set of genes and calculating the Euclidean distance between the replicates.

c. Functional analysis
We used the R package clusterProfiler (RRID:SCR_016884) to perform a Gene Ontology (GO) analysis (*Yu et al., 2012*) on the differentially expressed genes. GO terms with qvalue <0.05

were considered significant. GO annotations for *C. elegans* were obtained from R package org.Ce.eg.db: Genome wide annotation for Worm (*Carlson, 2018*).

 d. Data availability: RNA-seq data have been deposited and available at https://www.ncbi.nlm.nih.gov/bioproject/PRJNA576016

## RNA extraction and quantitative real-time PCR (qRT-PCR)

RNA was collected from day-1-adults, and embryos laid by 30–50 animals during 2–4 hr post heat shock. Adult animals were either passaged the previous day as L4s at densities of 20 worms/plate, or were bleach hatched (~100 eggs/plate). RNA extraction was conducted according to previously published methods (*Chikka et al., 2016*). Briefly, RNA samples were harvested in 50 µl of Trizol (catalog no. 400753, Life Technologies) and snap-frozen immediately in liquid nitrogen. For RNA extraction from embryos, the embryos were subjected to freeze-thaw cycles five times. The following steps were carried out immediately after snap-freezing or samples were stored at −80°C. Samples were thawed on ice and 200 µl of Trizol was added, followed by brief vortexing at room temperature. Samples were then vortexed at 4°C for at least 45 min to lyse the worms completely or lysed using a Precellys 24 homogenizer (Bertin Corp.) according to manufacturer's protocol. RNA was then purified as detailed in the manufacturer's protocol with appropriate volumes of reagents modified to 250 µl of Trizol. For RNA extraction from cultured cells and mouse cortical neurons, cells/neurons were washed with 1X PBS and then harvested in 800 µl of Trizol and snap-frozen in liquid nitrogen. RNA was extracted according to manufacturer's protocol with appropriate volumes of reagents modified to 800 µl of Trizol. The RNA pellet was dissolved in 17 µl of RNase-free water. The purified RNA was then treated with deoxyribonuclease using the TURBO DNA-free kit (catalog no. AM1907, Life Technologies) as per the manufacturer's protocol. In case of cultured cells and cortical neurons, 1 µg of total RNA was used for complementary DNA (cDNA) synthesis. cDNA was generated by using the iScript cDNA Synthesis Kit (catalog no. 170–8891, Bio-Rad). qRT-PCR was performed using LightCycler 480 SYBR Green I Master Mix (catalog no. 04887352001, Roche) in LightCycler 480 (Roche) or QuantStudio 3 Real-Time PCR System (Thermo Fisher Scientific) at a 10 µl sample volume, in a 96-well white plate (catalog no. 04729692001, Roche). The relative amounts of *hsp* mRNA were determined using the $\Delta\Delta C_t$ method (RRID:SCR_012155) for quantitation. Expression of GAPDH was used as internal control for samples obtained from NT2 cells and mouse primary cortical neurons. We selected *pmp-3* as an appropriate internal control for gene expression analysis in *C. elegans*. We and others have previously shown that *pmp-3* levels remain steady across numerous manipulations including stress and conditions that activate HSF-1 (*Ooi and Prahlad, 2017*; *White et al., 2012*; *Zhang et al., 2012*; *Kato et al., 2016*). We determined that indeed *pmp-3* expression levels remain stable upto heat-shock for 15 min at 34°C, when compared to two other internal controls widely used in the field (*act-1* and *gpd-3*) and when compared to a gene only expressed in adult animals (*syp*-1) and not embryos, avoiding the variability that could ensue from stochastic variation in the number of embryos in utero. The relative values of *pmp-3* compared to the three other internal control genes are shown in the table below.

***pmp-3* normalized to other internal controls**
**Fold change (Mean ± Standard Error) (n = 3)**

| Internal control | No heat shock | 15 min Heat Shock at 34°C | *P* value |
|---|---|---|---|
| *act-1* | 1 | 0.98 ± 0.03 | 0.53 |
| *syp-1* | 1 | 0.99 ± 0.01 | 0.29 |
| *gpd-3* | 1 | 1.00 ± 0.08 | 0.96 |

All relative changes of *hsp* mRNA were normalized to either that of the wild-type control or the control for each genotype (specified in figure legends). $\Delta\Delta C_t$ values were obtained in triplicate for each sample (technical replicates). Each experiment was then repeated a minimum of three times. For qPCR reactions, the amplification of a single product with no primer dimers was confirmed by melt-curve analysis performed at the end of the reaction. Reverse transcriptase-minus controls were included to exclude any possible genomic DNA amplification. Primers were designed using Roche's Universal Probe Library Assay Design Center software or Primer3 software (RRID:SCR_003139) and

generated by Integrated DNA Technologies. The primers used for the qRT-PCR analysis are listed below:

| Gene | Species | Forward primer (5'−3') | Reverse primer (5'−3') |
|---|---|---|---|
| hsp70 (C12C8.1) | C. elegans | TTGGTTGGGGGATCAACTCG | GAGCAGTTGAGGTCCTTCCC |
| hsp70 (F44E5.4/.5) | C. elegans | CTATCAGAATGGAAAGGTTGAG | TCTTTCCGTATCTGTGAATGCC |
| pmp-3 | C. elegans | TAGAGTCAAGGGTCGCAGTG | ATCGGCACCAAGGAAACTGG |
| syp-1 | C. elegans | GATGAAATGATAATTCGCCAAGA | ACGCAATCTTCCCTCATTTG |
| act-1 | C. elegans | ATCACCGCTCTTGCCCCATC | GGCCGGACTCGTCGTATTCTTG |
| gpd-3 | C. elegans | CAATGCTTCCTGCACCACTA | CTCCAGAGCTTTCCTGATGG |
| Hsp27 (Hspb1) | Mouse | ATCCCCTGAGGGCACACTTA | GGAATGGTGATCTCCGCTGAC |
| Hsp70 (Hspa1a) | Mouse | ATGGACAAGGCGCAGATCC | CTCCGACTTGTCCCCCAT |
| Cryab (Hspb5) | Mouse | CGGACTCTCAGAGATGCGTT | TGGGATCCGGTACTTCCTGT |
| Gapdh | Mouse | AACGACCCCTTCATTGAC | TCCACGACATACTCAGCAC |
| HSP70 (HSPA1A) | Human | CTACAAGGGGGAGACCAAGG | TTCACCAGCCTGTTGTCAAA |
| SPT16 (SUPT16H) | Human | GTGGAAAAGGCCATTGAAGA | GTGATAGCCCCAAAGTGCAT |
| GAPDH | Human | GAAGGTGAAGGTCGGAGTC | GAAGATGGTGATGGGATTTC |

## Western blotting

Western blot analysis was performed with adult day-1 animals. For protein analysis, 20–30 worms were harvested in 15 µl of 1X PBS (pH 7.4), and then 4X Laemmli sample buffer (catalog no. 1610737, Bio-Rad) supplemented with 10% β-mercaptoethanol was added to each sample before boiling for 30 min. Whole-worm lysates were resolved on 8% SDS-PAGE gels and transferred onto nitrocellulose membrane (catalog no. 1620115, Bio-Rad). Membranes were blocked with Odyssey Blocking Buffer (part no. 927–50000, LI-COR). Immunoblots were imaged using LI-COR Odyssey Infrared Imaging System (LI-COR Biotechnology, Lincoln, NE). Mouse anti-FLAG M2 antibody (catalog no. F1804, RRID:AB_262044, Sigma Aldrich) was used to detect HSF-1::FLAG. Rabbit anti-HA (catalog no. ab9110, RRID:AB_307019, Abcam) was used to detect HMG-3::HA. Mouse anti-α-tubulin primary antibody (AA4.3, RRID:AB_579793), developed by C. Walsh, was obtained from the Developmental Studies Hybridoma Bank (DSHB), created by the National Institute of Child Health and Human Development (NICHD) of the National Institute of Health (NIH), and maintained at the Department of Biology, University of Iowa. The following secondary antibodies were used: Sheep anti-mouse IgG (H and L) Antibody IRDye 800CW Conjugated (catalog no. 610-631-002, RRID:AB_220142, Rockland Immunochemicals) and Alexa Fluor 680 goat anti-rabbit IgG (H+L) (catalog no. A21109, RRID:AB_2535758, Thermo Fisher Scientific). LI-COR Image Studio software (RRID:SCR_015795) was used to quantify protein levels in different samples, relative to α-tubulin levels. Fold change of protein levels was calculated relative to wild-type/untreated controls.

For western blot analysis of mammalian cells, cells grown (1 × 10⁵ cells/ml density) in presence of dialyzed fetal bovine serum were washed with ice-cold 1X phosphate buffered saline (PBS), scrapped and pelleted by centrifugation at 300 g for 3 min at 4°C. Cell lysis was carried out using RIPA buffer (50 mM Tris (pH 7.4), 150 mM NaCl, 0.1% SDS, 1% NP-40, 0.5% sodium deoxycholate) supplemented with protease inhibitor cocktail (catalog no. 87785, Thermo Fisher Scientific). Protein concentration of whole-cell lysate was measured by Bradford assay (catalog no. 5000006, Bio-Rad) according to manufacturer's protocol. The preparation of samples, gel run, transfer of proteins to the membrane and imaging was performed as described earlier. Rabbit anti-HSF1 (catalog no. 4356, RRID:AB_2120258, Cell Signaling Technology) and rabbit anti-HSF1-S320 (catalog no. ab76183, RRID:AB_1523789, Abcam) were used to detect total and phosphorylated (S320) HSF1 respectively. Mouse anti-Hsp70 antibody (clone 3A3) was a gift from Dr. Richard I Morimoto, Northwestern University. Mouse anti-α-tubulin primary antibody (AA4.3, RRID:AB_579793) was used for detection of

tubulin which was used as internal control. Fold change of protein levels was calculated relative to wild-type/untreated controls.

## Chromatin immunoprecipitation (ChIP)

Preparation of samples for ChIP was performed by modifying the protocols previously described (*Li et al., 2016*; *Ooi and Prahlad, 2017*). Four hundred 1-day-old animals per condition (control or heat shock at 34°C for 5 or 15 min) were obtained by washing off two plates of bleach hatched animals, washed with 1X PBS (pH 7.4), and cross-linked with freshly prepared 2% formaldehyde (catalog no. 252549, Sigma Aldrich) at room temperature for 10 min. Reactions were quenched by adding 250 mM Tris (pH 7.4) at room temperature for 10 min and then washed three times in ice-cold 1X PBS supplemented with protease inhibitor cocktail and snap-frozen in liquid nitrogen. The worm pellet was resuspended in FA buffer [50 mM HEPES (pH 7.4), 150 mM NaCl, 50 mM EDTA, 1% Triton-X-100, 0.5% SDS and 0.1% sodium deoxycholate], supplemented with 1 mM DTT and protease inhibitor cocktail. We discovered during the course of experiments that the presence of a high concentration of EDTA was crucial for consistent yield of DNA, and to prevent the gradual degradation of DNA that otherwise occurred sporadically during the course of the experiments. We attribute this to the presence of resilient DNases that make their way into our preparation due to the culture condition of *C. elegans*. We assessed the quality of the DNA and ChIP with and without these higher concentrations of EDTA to ensure that the concentration of EDTA was not interfering with any other steps of ChIP. The suspended worm pellet was lysed using a Precellys 24 homogenizer (Bertin Corp.), and then sonicated in a Bioruptor Pico Sonication System (catalog no. B0106001, Diagenode) (15 cycles of 30 s on/off). All HSF-1 ChIP experiments were performed with wild-type (N2) and *tph-1* animals with FLAG tag at the C-terminus of the *hsf-1* gene. Anti-FLAG M2 magnetic bead (catalog no. M-8823, RRID:AB_2637089, Sigma-Aldrich) was used to immunoprecipitated endogenous HSF1. Beads were first pre-cleared with chromatin isolated from wild-type worms not having any FLAG tag and salmon sperm DNA (catalog no. 15632–011, Invitrogen). Worm lysate was incubated at 4°C overnight with the pre-cleared FLAG beads. For all other ChIP experiments, Protein A/G Magnetic Beads (catalog no. 88802, Pierce) pre-cleared with salmon sperm DNA was used. Precleared lysate was incubated at 4°C overnight with anti-RNA polymerase II (catalog no. 664906, RRID:AB_2565554, clone 8WG16, Bio legend), anti-Histone H3 (catalog no. ab1791, RRID:AB_302613, Abcam), anti-HA (catalog no. ab9110, RRID:AB_307019, Abcam) or control mouse (catalog no. sc-2025, RRID:AB_737182, Santa Cruz Biotechnology) and rabbit IgG antibody (catalog no. 2729, RRID:AB_1031062, Cell Signaling Technology) and then pre-cleared magnetic bead was added and incubated for another 3–4 hr. Beads were washed with low salt, high salt and LiCl wash buffers and then eluted in buffer containing EDTA, SDS and sodium bicarbonate (pH of the elution buffer was adjusted to 11). The elute was incubated with RNase A and then de-crosslinked overnight in presence of Proteinase K. The DNA was purified by ChIP DNA purification kit (catalog no. D5205, Zymo Research). qPCR analysis of DNA was performed as described above using primer sets specific for different regions of *hsp70* (*C12C8.1*) and *hsp70* (*F44E5.4/.5*) genes. The primer pair used for amplifying the promoter region of *hsp70* (*C12C8.1*) gene immunoprecipitated by FLAG beads (for HSF-1 ChIP) was not suitable to amplify DNA immunoprecipitated by RNA polymerase II. Therefore, we used a different primer pair that recognizes slightly downstream region of *hsp70* (*C12C8.1*) gene for RNA polymerase II ChIP as mentioned in the table below and also in the figure legends. Promoter region of *syp-1* was amplified for all HSF1-ChIP experiments to quantify non-specific binding of HSF-1 (*Figure 4—figure supplement 1C*). Chromatin immunoprecipitated by all primary antibodies were compared with corresponding rabbit or mouse control IgG to confirm the specificity (*Figure 4—figure supplement 2*, *Figure 5—figure supplement 2*). For all ChIP experiments, 10% of total lysate was used as 'input' and chromatin immunoprecipitated by different antibodies were expressed as % input values. All relative changes were normalized to either that of the wild-type control or the control of each genotype (specified in figure legends) and fold changes were calculated by $\Delta\Delta C_t$ method. The primers used for ChIP experiments, and the expected amplicon sizes are as follows:

| Gene name | Position | Forward primer (5′—3′) | Reverse primer (5′—3′) | Amplicon size | Antibody used |
|---|---|---|---|---|---|
| C12C8.1 | Promoter | CTCAGGCAGTG GAAGAACTAAA | TTATACGTTCCTC TGGCATCTTC | 88 bp | FLAG (HSF1-FLAG), H3, HA (HMG-3-HA) |
| F44E5.4/.5 | Promoter | ATACTACCCGAATCCCAGCC | GCAACAGAGACGCAGATTGT | 150 bp | FLAG (HSF1-FLAG), H3, HA (HMG-3-HA), RNA polymerase II |
| C12C8.1 | Region A | ATCGACTTGGGTA CTACGTACTC | CTTGTTCCCTTCGGAGTTCG | 161 bp | RNA polymerase II |
| syp-1 | Promoter | CAACAAAACGCGCTCCATT | GGAGGCCGCAAACACC | 80 bp | FLAG (HSF1-FLAG) |
| C12C8.1 | Region B | TGTACTTGGGCA TTCTGTACGG | GCATTGAGTCCAG CAATAGTAGC | 108 bp | RNA polymerase II, H3 |
| C12C8.1 | Region C | ACAATTCGCAAT GAGAAGGGACG | GCATCTTCTGCT GATAACAGTGATC | 191 bp | RNA polymerase II |
| F44E5.4/.5 | Region B | TGATCTTCGATCTCGGAGGAGG | TCACAAGCAGTTCGGAGACG | 220 bp | RNA polymerase II, H3 |
| F44E5.4/.5 | Region C | TTGATGAAACACTTCGTTGGTTGG | TCCAGCAGTTCCAGGATTTC | 170 bp | RNA polymerase II |
| C12C8.1 | TSS | ACGTACTCATGTGTCGGTAT | TCTTCTTCCAGTTTACATAATCCT | 92 bp | H3, HA (HMG-3-HA) |
| F44E5.4/.5 | TSS | TAAAAGGGCTGGGATTCGGG | ACCGAGGTCGATACCAATAGC | 118 bp | H3, HA (HMG-3-HA) |
| C12C8.1 | TSS-50 | AACTCAAATCTTATGCAGAAT | CGTAGTACCCAAGTCGATTCCA | 119 bp | HA (HMG-3-HA) |
| F44E5.4/.5 | TSS-50 | GTCGGCCGTCTCTTTCTCTT | CCCGAATCCCAGCCCTTTT | 157 bp | HA (HMG-3-HA) |

## Statistical analysis

Each 'experiment' refers to a biological repeat. No statistical methods were used to predetermine sample size. The experiments were not randomized. A minimum of three independent experiments (starting from independent parent populations of *C. elegans*) were conducted for all data points. However, many experiments were repeated in multiple contexts and n numbers, and mean values reflect all repeats. All qPCR experiments in *C. elegans* were conducted on 30–200 animals per experiment. All ChIP-qPCR experiments were conducted on 400 animals per sample per experiment. For pairwise comparisons such as for qRT-PCR data and progeny hatching, significance was tested using Paired Student's t tests (assumptions of parametric distributions were first tested and were fulfilled). For all ChIP-qPCR experiments where multiple comparisons were made, significance was tested using one-way ANOVA with Tukey's multiple comparison correction. Data are indicated as mean ± standard error. p values are indicated as follows: $^*p<0.05$, $^{**}p<0.01$, $^{***}p<0.001$. FDR calculations for the RNAseq data set are described in the RNA-seq section of the Methods.

## Acknowledgements

We thank the members of VP laboratory, Dr. Sarit Smolikove, Dr. Bin He, Dr. Chris Stipp and Dr. Tali Gidalevitz for their helpful comments, Kat Dvorak, Matthew Wheat, Dr. Rachel Reichman and Gery Hehman for technical help, and Dr. Smolikove for advice with CRISPR/Cas9. The BAT1560 strain was a kind gift from Dr. Baris Tursun, Max Delbrück Center (MDC). We also thank the anonymous reviewers of a prior submission for extremely helpful comments. HSP70 antibodies were a kind gift of Dr. Morimoto (Northwestern University). Nematode strains were provided by the Caenorhabditis Genetics Center (CGC) (funded by the NIH Infrastructure Programs P40 OD010440). JAW and LCF are supported by R01 NS055272 to JAW. This work was supported solely by the Aging Mind and Brian Initiative (AMBI), University of Iowa (VP) and by NIH R01 AG 050653 (VP).

## Additional information

### Funding

| Funder | Grant reference number | Author |
|---|---|---|
| University of Iowa | Aging Mind and Brain Initiative | Veena Prahlad |

| National Institutes of Health | AG 050653 | Veena Prahlad |

The funders had no role in study design, data collection and interpretation, or the decision to submit the work for publication.

## Author contributions

Srijit Das, Data curation, Formal analysis, Investigation, Methodology, Writing - original draft, Writing - review and editing; Felicia K Ooi, Leah C Fuller, Joshua A Weiner, Methodology; Johnny Cruz Corchado, Data curation, Formal analysis, Writing - review and editing; Veena Prahlad, Conceptualization, Resources, Formal analysis, Supervision, Funding acquisition, Methodology, Writing - original draft, Project administration, Writing - review and editing

## Author ORCIDs

Srijit Das [ID] https://orcid.org/0000-0002-0402-3256
Felicia K Ooi [ID] http://orcid.org/0000-0003-4766-2477
Joshua A Weiner [ID] http://orcid.org/0000-0002-3352-2847
Veena Prahlad [ID] https://orcid.org/0000-0002-0413-6074

## Decision letter and Author response

Decision letter https://doi.org/10.7554/eLife.55246.sa1
Author response https://doi.org/10.7554/eLife.55246.sa2

# Additional files

## Supplementary files

• Supplementary file 1. Genes differentially expressed (FDR < 0.01) in wild-type animals upon 5 min heat shock at 34°C. Day one adult wild-type N2 animals were exposed to 34°C for 5 min. Animals were harvested immediately, and differential gene expression (FDR < 0.01) was calculated by comparing heat-shocked animal with control animals at 20°C.

• Supplementary file 2. No differential expression (FDR < 0.01) could be detected in *hsf-1* mutant animals *hsf-1(sy441) I* upon 5 min heat shock at 34°C. Day one adult *hsf-1 (sy441) I* animals were exposed to 34°C for 5 min. Animals were harvested immediately, and differential gene expression (FDR < 0.01) was calculated by comparing heat-shocked animal with control *hsf-1 (sy441) I* animals at 20°C. Note: no differential gene expression was detected.

• Supplementary file 3. Genes differentially expressed (FDR < 0.01) in *tph-1 (mg280) II* animals upon 5 min heat shock at 34°C. Day one adult *tph-1 (mg280) II* animals were exposed to 34°C for 5 min. Animals were harvested immediately, and differential gene expression (FDR < 0.01) was calculated by comparing heat-shocked animal with control *tph-1 (mg280) II* animals at 20°C.

• Transparent reporting form

## Data availability

RNA-seq data have been deposited and available at https://www.ncbi.nlm.nih.gov/bioproject/PRJNA576016.

The following datasets were generated:

| Author(s) | Year | Dataset title | Dataset URL | Database and Identifier |
| --- | --- | --- | --- | --- |
| Das S, Ooi FK, Cruz-Corchado J, Fuller LC, Weiner JA, Prahlad V | 2019 | Genes deferentially expressed in wild type C. elegans, tph-1 mutants and hsf1(sy441) mutants upon a transient heat shock. | https://www.ncbi.nlm.nih.gov/bioproject/PRJNA576016 | NCBI BioProject, PRJNA576016 |
| Cruz-Corcahdo J, Ooi FK, Das S, Prahlad V | 2019 | Transcriptome of C. elegans upon alteration of 5-HT | https://www.ncbi.nlm.nih.gov/bioproject/PRJNA594152 | NCBI BioProject, PRJNA594152 |

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
