## [Decision Letter]

**Acceptance summary:**

This study revealed that in *C. elegans*, maternal neurons release serotonin that induces a rapid heat shock response in the germ cells, thus ensuring viability of the offspring and their subsequent heat resilience. The conceptual advances include that heat shock response can be rapid in a whole organism, involves inter-tissue communication, with trans-generational benefits.

**Decision letter after peer review:**

Thank you for submitting your article "Serotonin signaling by maternal neurons upon stress ensures progeny survival" for consideration by *eLife*. Your article has been reviewed by three peer reviewers, and the evaluation has been overseen by a Reviewing Editor and Jessica Tyler as the Senior Editor. The following individual involved in review of your submission has agreed to reveal their identity: Michael Petrascheck (Reviewer #2).

The reviewers have discussed the reviews with one another and the Reviewing Editor has drafted this decision to help you prepare a revised submission. In recognition of the fact that revisions may take longer than we typically allow, until the research enterprise restarts in full, we will give authors as much time as they need to submit revised manuscripts.

Summary:

The manuscript revealed that in *C. elegans*, maternal neurons release serotonin that induces a rapid heat shock response in the germ cells, thus ensuring viability of the offspring and their subsequent heat resilience.

Essential revisions:

The reviewers in general are pleased with the conceptual advances of the findings and the high quality of the data. While a number of worthwhile experiments have been suggested, in consultation with the reviewers, the general sentiment is the manuscript is already extensive and data rich and should be published without much further delay.

Some technical points that should be addressed:

1) What is the epistasis between *tph-1* and *hsf-1*? That is, what happens to viability upon heat shock when they are removed in combination? Figure 1.

2) The experiments that use qPCR to measure chaperone levels appear to rely on only one internal control gene, *pmp-3*. Using just one gene, rather than a geometric mean of multiple genes, can create misleading results if the expression levels of that gene are altered under the conditions used (heat stress, for example). If they are going to use this single control gene, do the authors have evidence that *pmp-3* expression levels do not substantially change upon heat stress, *hsf-1* knockdown, etc?

Reviewer #1:

The main question of the manuscript by Das et al. is 'whether and how sensory information used by the organism to predict impending danger is coupled to the protection of its future offspring'. The authors have made the interesting observation that a 5 min heat shock of *C. elegans* mothers promotes progeny survival in the next generation. This effect occurs via serotonin signaling from the mothers, which activates HSF1 in the germline. They show that the histone chaperone FACT is involved in this process, and that in cultured mammalian cells, serotonin phosphorylates PKA which in turn recruits HSF1. Some aspects of the work are rigorous, however the following points would need to be addressed to justify the claim that the authors have "elucidated a molecular mechanism by which transcriptional response times of specific cells are tuned to stimulus intensity and onset". Additionally there are some open questions regarding the mechanism of serotonin action in *C. elegans* that must be addressed.

1) One of the arguments made in the paper is that serotonin is a stress signal. That is partially true. There is an entire body of evidence, from many different labs, that it is also an indicator of food availability. The different serotonergic neurons (ADF, NSM, HSM) play different roles in sensing stress, food availability, coordinating reproduction to the availability of food, and actual food entering the pharynx. For their phenotypes, the authors would need to conduct necessity and sufficiency experiments (selective *tph-1* depletion and rescue) to identify which serotonergic neurons drive their germline phenotype. The answer to this question is critical, without which the relevance of the 5 min 34 degree heat shock remains unclear (more on this below), and is experimentally feasible, and is a standard in the field.

2) What is the epistasis between *tph-1* and *hsf-1*? That is, what happens to viability upon heat shock when they are removed in combination? Figure 1.

3) It is also somewhat puzzling that loss of either *tph-1* or *hsf-1* still retains 50% viability (Figure 1). What does this mean? That this pathway is one factor amongst others in every germline cell that play an equally relevant role, or that half the embryos don't require either *tph-1* or *hsf-1*? It would be good to resolve these alternative hypotheses experimentally. Perhaps ChIP seq (rather than ChIP qPCR) would address this question, but other approaches would also be valid as long as they are experimental.

4) What does it mean for this particular type of heat shock response to be present at 5 minutes, but gone at 15 minutes? While potentially interesting, the significance of this is totally unclear. Without addressing this point, the relevance of the experimental manipulation comes into question.

5) In the worm, which serotonin receptor/s is involved? The authors imply (the heterozygote experiment is clever) that a cell-non-autonomous mechanism is at play. Maternal serotonin neurons control an HSF1 mediated germline protective response. There are 2 possibilities: direct (i.e. there is a serotonin receptor on the germline cells that detects neuronally released serotonin), and indirect (serotonin works on a receptor somewhere else, which triggers a systemic or germline-specific response). This fundamental question can be experimentally addressed in a variety of tractable ways using tools and resources that are available. Without a deeper investigation into how serotonin is working, the relevance and significance are again thrown into question.

6) There are a few typographical errors throughout.

Reviewer #2:

Das and colleagues describe a new, evolutionary conserved role of serotonin signaling in which neuronal signals act to protect future offspring from heat stress. They show that serotonin signals elicited by stress control transcription in germ cells by activating HSF-1, which in turn recruits the histone chaperone FACT. This cascade induces transcription of protective genes that increase embryo survival but only if it was exposed to maternal serotonin to activate the protective response. This is a very interesting story and rigorously done. I enjoyed reading it.

Reviewer #3:

This work identifies a mechanism by which short term heat shock influences germline transcription, resulting in increased embryonic viability and transgenerational increase in thermotolerance. This mechanism seems to operate through the release of serotonin, leading to activation of PKA and HSF-1 in the germline, and the recruitment of a component of the histone chaperone FACT complex, which results in histone displacement. The study also demonstrates the conservation of some of these findings, showing that in mammalian neurons serotonin can also induce transcriptional up regulation of chaperone genes.

Overall, the range of techniques and approaches used is impressive, and the story presented is complex and extensive. The idea of a mechanism that acts so swiftly to protect the germline upon heat stress is very interesting – and, in fact, very surprising, given the transient (5 minute) nature of the stress, and the fact that animals in such a short duration of elevated temperature are likely to undergo only a relatively small increase in temperature themselves. However, the authors provide a large amount of experimental evidence to support this idea.

Given the large amount of data provided I don't think it is appropriate to suggest many extra experiments. There were a few more substantial issues, however, that are worth raising here.

1) The experiments that use qPCR to measure chaperone levels appear to rely on only one internal control gene, *pmp-3*. Using just one gene, rather than a geometric mean of multiple genes, can create misleading results if the expression levels of that gene are altered under the conditions used (heat stress, for example). If they are going to use this single control gene, do the authors have evidence that *pmp-3* expression levels do not substantially change upon heat stress, *hsf-1* knockdown, etc?

2) The mammalian cell data relies upon the use of inhibitors to suppress the activity of serotonin receptors, and PKA. Given that inhibitors frequently have non-specific effects, this data would be more convincing if confirmed using siRNA against the 5-HT4 receptor and PKA.

---

## [Author Response]

Essential revisions:The reviewers in general are pleased with the conceptual advances of the findings and the high quality of the data. While a number of worthwhile experiments have been suggested, in consultation with the reviewers, the general sentiment is the manuscript is already extensive and data rich and should be published without much further delay.Some technical points that should be addressed:1) What is the epistasis between tph-1 and hsf-1? That is, what happens to viability upon heat shock when they are removed in combination? Figure 1.2) The experiments that use qPCR to measure chaperone levels appear to rely on only one internal control gene, pmp-3. Using just one gene, rather than a geometric mean of multiple genes, can create misleading results if the expression levels of that gene are altered under the conditions used (heat stress, for example). If they are going to use this single control gene, do the authors have evidence that pmp-3 expression levels do not substantially change upon heat stress, hsf-1 knockdown, etc?Reviewer #1:The main question of the manuscript by Das et al. is 'whether and how sensory information used by the organism to predict impending danger is coupled to the protection of its future offspring'. The authors have made the interesting observation that a 5 min heat shock of C. elegans mothers promotes progeny survival in the next generation. This effect occurs via serotonin signaling from the mothers, which activates HSF1 in the germline. They show that the histone chaperone FACT is involved in this process, and that in cultured mammalian cells, serotonin phosphorylates PKA which in turn recruits HSF1. Some aspects of the work are rigorous, however the following points would need to be addressed to justify the claim that the authors have "elucidated a molecular mechanism by which transcriptional response times of specific cells are tuned to stimulus intensity and onset". Additionally there are some open questions regarding the mechanism of serotonin action in C. elegans that must be addressed.1) One of the arguments made in the paper is that serotonin is a stress signal. That is partially true. There is an entire body of evidence, from many different labs, that it is also an indicator of food availability. The different serotonergic neurons (ADF, NSM, HSM) play different roles in sensing stress, food availability, coordinating reproduction to the availability of food, and actual food entering the pharynx. For their phenotypes, the authors would need to conduct necessity and sufficiency experiments (selective tph-1 depletion and rescue) to identify which serotonergic neurons drive their germline phenotype. The answer to this question is critical, without which the relevance of the 5 min 34 degree heat shock remains unclear (more on this below), and is experimentally feasible, and is a standard in the field.

We appreciate the reviewer’s comments and agree, wholeheartedly, that how serotonin acts to control stress versus how it controls feeding and foraging remains an outstanding question in the field. The optogenetics experiments, and our previous publication (Tatum et al., 2015) address, in part, the question of which neurons are involved in stress regulation: ChR2 is expressed under a modified *tph-1* promoter only in the ADF and NSM neurons and optogenetically exciting these neurons is sufficient to activate HSF-1 in a manner dependent on FACT (Figure 6—figure supplement 1). However, these are the same neurons involved in feeding and foraging, implying that the separation of the two functions may be more complex and not simply a function of which neurons are active. The question is part of our ongoing research, but we feel answering this is beyond the scope of the current studies. We have included a sentence in the manuscript to clarify which neurons are excited in the optogenetics experiment (subsection “Serotonin-dependent recruitment of FACT to displace histones hastens the onset of transcription”).

2) What is the epistasis between tph-1 and hsf-1? That is, what happens to viability upon heat shock when they are removed in combination? Figure 1.

We have included experiments to address this important question, and show that *hsf-1* knock-down does not exacerbate/change the effects of loss of 5-HT, suggesting that the two are part of the same genetic pathway (Figure 1A).

3) It is also somewhat puzzling that loss of either tph-1 or hsf-1 still retains 50% viability (Figure 1). What does this mean? That this pathway is one factor amongst others in every germline cell that play an equally relevant role, or that half the embryos don't require either tph-1 or hsf-1? It would be good to resolve these alternative hypotheses experimentally. Perhaps ChIP seq (rather than ChIP qPCR) would address this question, but other approaches would also be valid as long as they are experimental.

The reviewer makes a good point and it is possible that serotonin release is one amongst other redundant pathways that protect progeny. We are currently investigating why and when embryos die, but the inclusion of more details, we think, is beyond the scope of this study. Our data also show that it isn’t that 50% of the *tph-1* embryos require 5-HT or HSF-1: more die with lengthier heat exposure. In addition, germ cells in wild-type animals are also susceptible to heat stress, if the heat-exposure extends to 15 minutes. The difference is that in wild-type animal, a longer exposure of mothers to heat stress is required to cause embryonic death at numbers comparable to that in *tph-1* mutants. Upon 5 min of heat shock of parents, viability of post-HS fertilized wild-type embryos=94 ± 2 % while that of *tph-1*=53 ± 2%. Upon 15 min of heat shock of parents: N2 viability=33 ± 5 % and *tph-1* viability=9 ± 1.5 %.

4) What does it mean for this particular type of heat shock response to be present at 5 minutes, but gone at 15 minutes? While potentially interesting, the significance of this is totally unclear. Without addressing this point, the relevance of the experimental manipulation comes into question.

We thank the reviewer for an opportunity to clarify this point. We have now included a sentence in the Discussion explicitly highlighting the existence of kinetic changes during the course of a heat exposure of cells and animals (which these data as well as other published studies in the field support). Our interpretation of our data is that during 5 minutes, 5-HT accelerates the onset of HSF-1 dependent transcription. We detect this as a difference in expression levels of HSF-1 transcription, the recruitment of a subunit of the FACT complex to the transcribed genes, germ cell protection and an intergenerational increase in resilience to heat stress. This ‘quick-start’, so to speak, that is mediated by perception of the environment allows the animal to better protect its progeny from the detrimental effects of heat. Without serotonin the animal still is able to activate HSF-1 (by 15 minutes), but because this activation occurs later it is unable to mitigate the detrimental effects of heat stress that, likely, have already begun to occur.

5) In the worm, which serotonin receptor/s is involved? The authors imply (the heterozygote experiment is clever) that a cell-non-autonomous mechanism is at play. Maternal serotonin neurons control an HSF1 mediated germline protective response. There are 2 possibilities: direct (i.e. there is a serotonin receptor on the germline cells that detects neuronally released serotonin), and indirect (serotonin works on a receptor somewhere else, which triggers a systemic or germline-specific response). This fundamental question can be experimentally addressed in a variety of tractable ways using tools and resources that are available. Without a deeper investigation into how serotonin is working, the relevance and significance are again thrown into question.

Our experiments do not distinguish between the two possibilities. We have now added a sentence explicitly stating that (subsection “Serotonin-induced PKA-activation is a conserved signaling pathway that enables HSF1 to recruit FACT”). We had shown (Tatum et al., 2015 ) that some of the effects of serotonin on HSF1 in the gonad occur through 5-HT2 and we have unpublished RNA smFISH data to show that 5-HT2 mRNA is present in gonadal sheath cells. However, 5HT2 does not act through cAMP/ PKA. Therefore more needs to be done to understand the distribution and activity of 5-HT receptors in *C. elegans* somatic tissues. We think the best way to do this is by tagging endogenous receptors, for obvious functional reasons, but also because using different promoter lengths to express reporters has shown different expression patterns in vivo. We intend to pursue this question in a rigorous manner in the future.

6) There are a few typographical errors throughout.

Thank you. We believe we have corrected them.

Reviewer #2:Das and colleagues describe a new, evolutionary conserved role of serotonin signaling in which neuronal signals act to protect future offspring from heat stress. They show that serotonin signals elicited by stress control transcription in germ cells by activating HSF-1, which in turn recruits the histone chaperone FACT. This cascade induces transcription of protective genes that increase embryo survival but only if it was exposed to maternal serotonin to activate the protective response. This is a very interesting story and rigorously done. I enjoyed reading.Reviewer #3:This work identifies a mechanism by which short term heat shock influences germline transcription, resulting in increased embryonic viability and transgenerational increase in thermotolerance. This mechanism seems to operate through the release of serotonin, leading to activation of PKA and HSF-1 in the germline, and the recruitment of a component of the histone chaperone FACT complex, which results in histone displacement. The study also demonstrates the conservation of some of these findings, showing that in mammalian neurons serotonin can also induce transcriptional up regulation of chaperone genes.Overall, the range of techniques and approaches used is impressive, and the story presented is complex and extensive. The idea of a mechanism that acts so swiftly to protect the germline upon heat stress is very interesting – and, in fact, very surprising, given the transient (5 minute) nature of the stress, and the fact that animals in such a short duration of elevated temperature are likely to undergo only a relatively small increase in temperature themselves. However, the authors provide a large amount of experimental evidence to support this idea.Given the large amount of data provided I don't think it is appropriate to suggest many extra experiments. There were a few more substantial issues, however, that are worth raising here.1) The experiments that use qPCR to measure chaperone levels appear to rely on only one internal control gene, pmp-3. Using just one gene, rather than a geometric mean of multiple genes, can create misleading results if the expression levels of that gene are altered under the conditions used (heat stress, for example). If they are going to use this single control gene, do the authors have evidence that pmp-3 expression levels do not substantially change upon heat stress, hsf-1 knockdown, etc?

Prior to these studies our lab had confronted this problem because (a) we wanted to ensure that the expression levels of the internal control does not change with heat shock, (b) we were worried that the numbers of embryos in utero would vary stochastically (especially upon heat shock), and knowing that hsps and other inducible genes are not expressed during early embryogenesis, but an internal control such as actin would be detectable, we were worried that the choice of the internal control could aberrantly report on relative changes and (c) we wanted to match the efficiency of PCR amplification between hsps (which, despite being highly expressed upon stress, amplify with Ct values lower than many of the commonly used internal controls such as actin). This led to our choice of four different internal controls, *pmp-3* which was reported to remain steady across numerous manipulations, *syp-1*, a gene that encodes a synaptonemal complex protein and therefore is expressed only in the adult and not in embryos, and actin and *gpd-3*, which are more constitutively expressed and commonly used. We include data showing that *pmp-3* levels do not change upon heat-stress relative to *syp-1*, actin and *gpd-3*. We therefore chose to use *pmp-3* for all our studies. We have included data in the Materials and methods section showing that *pmp-3* levels do not change with heat-shock with respect to three other internal controls (subsection “RNA extraction and quantitative real-time PCR (qRT-PCR”).

2) The mammalian cell data relies upon the use of inhibitors to suppress the activity of serotonin receptors, and PKA. Given that inhibitors frequently have non-specific effects, this data would be more convincing if confirmed using siRNA against the 5-HT4 receptor and PKA.

In the course of our experiments we saw that a 24 hour knock-down of PKA was detrimental to cell survival and growth, and that is why we opted to use a fast-acting chemical inhibitor, applied transiently. The PKA inhibitor used, H89, has been used in previous published studies on HSF1 to demonstrate that the S320 phosphorylation site on HSF1 is because of PKA activity. We show in our experiments that treatment with H89 decreases S320 phosphorylation, and therefore is acting as expected.